# Capacity-Aware Inference: Mitigating the Straggler Effect in Mixture of Experts

**Shwai He**[1]    **Weilin Cai**[2]    **Jiayi Huang**[2]    **Ang Li**[1]

[1]University of Maryland, College Park
[2]The Hong Kong University of Science and Technology (Guangzhou)
{shwaihe, angliece}@umd.edu
wcai738@connect.hkust-gz.edu.cn, hjy@hkust-gz.edu.cn

## Abstract

The Mixture of Experts (MoE) is an effective architecture for scaling large language models by leveraging sparse expert activation to balance performance and efficiency. However, under expert parallelism, MoE suffers from inference inefficiencies due to imbalanced token-to-expert assignment, where underloaded experts complete computations early but must wait for overloaded experts, leading to global delays. We define this phenomenon as the ***Straggler Effect***, as the most burdened experts dictate the overall inference latency. To address this, we first propose ***Capacity-Aware Token Drop***, which enforces expert capacity limits by discarding excess tokens from overloaded experts, effectively reducing load imbalance with minimal performance impact (e.g., 30% speedup with only 0.9% degradation on OLMoE). Next, given the presence of low-load experts remaining well below the capacity threshold, we introduce ***Capacity-Aware Expanded Drop***, which allows tokens to include additional local experts in their candidate set before enforcing strict local capacity constraints, thereby improving load balance and enhancing the utilization of underused experts. Extensive experiments on both language and multimodal MoE models demonstrate the effectiveness of our approach, yielding substantial gains in expert utilization, model performance, and inference efficiency, e.g., applying Expanded Drop to Mixtral-8×7B-Instruct yields a 0.2% average performance improvement and a 1.85× inference speedup. The code is released at: https://github.com/CASE-Lab-UMD/Capacity-Aware-MoE.

## 1 Introduction

In recent years, the rapid evolution of Large Language Models (LLMs) (OpenAI, 2024; Team, 2024a; et al., 2024b) has driven a wave of innovations, continuously expanding the frontiers of AI research and applications. Among the model architectural innovations, the Mixture of Experts (MoE) framework has emerged as a pivotal technique for optimizing the cost-performance trade-off in LLMs. Specifically, MoE (Shazeer et al., 2017b) enhances scalability by integrating multiple experts while activating only a subset per input. This selective activation substantially improves model performance without a corresponding increase in computational cost, effectively balancing efficiency and performance.

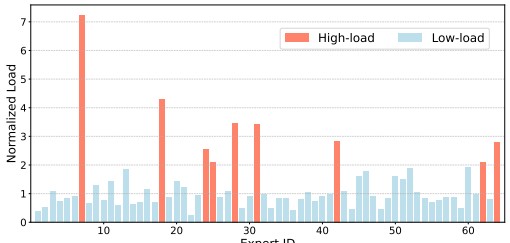

Figure 1: **Illustration of the Straggler Effect in MoE Inference.** The normalized load is computed as each expert's load divided by the mean load across all experts. Example shown with OLMoE (Muennighoff et al., 2024) on OpenBookQA (Mihaylov et al., 2018b).

Despite the success of MoE, a key efficiency challenge lies in the imbalanced token-to-expert distribution, which results in some experts being overloaded while others remain underutilized (Lepikhin et al., 2021; Zoph et al., 2022). In distributed GPU settings, experts are typically sharded across multiple devices, with each GPU responsible for a subset of the experts. Under expert parallelism, low-load experts complete their computations earlier but must wait for overloaded experts

to finish, as synchronization barriers are required before proceeding to the next stage. This expert-level straggler effect further propagates to device-level delays, where GPUs hosting lighter expert workloads are stalled by GPUs hosting heavier workloads, leading to inefficient resource utilization and increased end-to-end latency during inference. As illustrated in Figure 1, this phenomenon is referred to as the ***Straggler Effect***, where the heavily loaded experts determine the overall latency of imbalanced MoE inference.

While auxiliary balance losses have been incorporated into the training process to alleviate imbalance (Shazeer et al., 2017b; Fedus et al., 2022; et al., 2024b), these techniques remain ineffective in mitigating imbalance during inference. Specifically, as shown in Figure 1, our findings reveal a highly uneven token distribution among experts, with the highest-load expert handling more than seven times the expected average load. Moreover, managing such imbalance during inference often incurs additional resource overhead. For example, DeepSeek-V3 mitigates this issue by duplicating high-load experts and deploying them redundantly across devices (et al., 2024b). This motivates us to explore *efficient token-to-expert assignment* by addressing the key question: *How can we prevent extreme overloading of heavily utilized experts?*

We propose ***Capacity-Aware Inference*** to address this challenge. Specifically, for high-load experts, we introduce ***Capacity-Aware Token Drop***, which imposes a maximum capacity constraint and discards excess tokens from overloaded experts. This approach alleviates severe load imbalance and significantly improves efficiency, while maintaining model performance since the dropped tokens represent only a small fraction of the total workload, e.g., OLMoE achieves a 30% speedup in MoE layers with just a 0.9% performance degradation. After removing excess tokens from overloaded experts, we observe that some low-load experts remain significantly underutilized relative to the predefined capacity constraints, yet must still wait for other experts to complete their computations. This leads us to a second key question: *How can we effectively leverage the available capacity of underutilized experts?*

For low-load experts, we extend Token Drop with ***Capacity-Aware Expanded Drop***, which further utilizes the available capacity of underutilized experts to handle overflow tokens from high-load experts. Specifically, under expert parallelism across multiple GPUs, Expanded Drop allows each token to consider additional candidate experts on the same device while still enforcing strict local capacity constraints. This expanded selection improves the utilization of low-load experts and enhances the representational capacity of MoE models under capacity-constrained scenarios.

Extensive experimental results validate the effectiveness of our proposed techniques, demonstrating significant improvements in both efficiency and performance, e.g., *applying Expanded Drop to Mixtral-8×7B-Instruct yields a 0.2% average performance improvement and a 1.85× inference speedup*. Moreover, in multimodal models, we identify redundancy among image tokens and show that applying aggressive capacity constraints (e.g., setting the maximum to half of the average expert load) can still maintain performance. In short, our contributions are four-fold:

- We explicitly clarify and analyze the Straggler Effect caused by test-time token imbalance in the Mixture of Experts, highlighting the optimization potential for reducing latency.
- Toward the high-load experts, we propose **Capacity-Aware Token Drop**, which enforces capacity constraints by discarding excess tokens assigned to overloaded experts, thereby mitigating extreme load imbalance.
- To better utilize underloaded experts, we introduce **Capacity-Aware Expanded Drop**, which expands the candidate expert set to include additional local experts, further improving load balance and model performance.
- Extensive experiments on both language and multimodal models validate the effectiveness of our approach, demonstrating substantial improvements in inference efficiency with comparable performance.

## 2 RELATED WORKS

**Mixture of Experts Models**  The Mixture of Experts (MoE) is a neural network architecture with an extended set of parameters (referred to as "experts") controlled by a router, and was first introduced in the context of conditional computation (Jacobs et al., 1991; Jordan & Jacobs, 1994). The potential of sparse activation in MoE is subsequently exploited by Shazeer et al. (2017a) for efficient training

and inference on pretrained models with special designs, opening the door for MoE in various vision (Riquelme et al., 2021) and language (Lepikhin et al., 2020; Du et al., 2022; Fedus et al., 2022) scenarios. Due to its exceptional efficiency, MoE has been adopted as a foundational framework in the designs of large language models (Jiang et al., 2024; Dai et al., 2024; Xue et al., 2024a; Zhu et al., 2024; Team, 2024b), achieving superior scaling laws at low computational costs. Despite these advancements, MoE still faces efficiency challenges in both training and inference (Cai et al., 2024), and our work specifically focuses on enhancing inference-time efficiency.

**Imbalance in Mixture of Experts**  The imbalance in token-to-expert assignments (Zhou et al., 2022; Chen et al., 2022) poses a significant challenge to the deployment of MoE. This imbalance leads to inefficiencies in computation, communication, and memory (He et al., 2023; Song et al., 2023; Xue et al., 2024b), making it a critical bottleneck for MoE scalability and deployment. To mitigate this issue, an auxiliary balance loss (Shazeer et al., 2017b) is incorporated into the training process to encourage more uniform token distribution across experts. Additionally, various training strategies have been introduced to further balance token assignments: Switch-Transformer (Fedus et al., 2022) and DeepSeek-V2 (et al., 2024a) implement Token Drop to alleviate expert overload, while DeepSeek-V3 (et al., 2024b) introduces an additional sequence-level auxiliary loss to prevent severe token imbalance.

However, these techniques primarily focus on training and fail to ensure balanced token assignments during inference. Instead, addressing token imbalance at inference often incurs additional resource costs. For example, DeepSeek-V3 (et al., 2024b) mitigates this issue by duplicating high-load experts and deploying them redundantly. In contrast, our approach effectively balances token assignments without introducing additional computational overhead.

## 3 BACKGROUND AND MOTIVATION

### 3.1 EXTREMELY IMBALANCED EXPERT UTILIZATION

A Mixture of Experts (MoE) layer consists of a collection of $n$ experts, $\{\boldsymbol{E}_1, \boldsymbol{E}_2, \ldots, \boldsymbol{E}_n\}$ and a router $\boldsymbol{G}$ that dynamically selects the most relevant experts for a given input $\boldsymbol{x}$. The router computes selection scores $\boldsymbol{G}(\boldsymbol{x})$, for all experts and selects the top $k$ experts, resulting in a sparse activation:

$$\mathcal{K} = \mathrm{TopK}(\mathrm{Softmax}(\boldsymbol{G}(\boldsymbol{x})), k). \tag{1}$$

The input $\boldsymbol{x}$ is processed by the selected experts, and their outputs are combined into a weighted sum based on the router's scores. This process is mathematically expressed as:

$$\boldsymbol{y} = \sum\nolimits_{i \in \mathcal{K}} \boldsymbol{G}(\boldsymbol{x})_i \cdot \boldsymbol{E}_i(\boldsymbol{x}), \tag{2}$$

where $\mathcal{K}$ denotes the indices of selected experts, $\boldsymbol{G}(\boldsymbol{x})_i$ represents the selection score for the $i$-th expert, and $\boldsymbol{E}_i(\boldsymbol{x})$ is the output from the $i$-th expert. In transformer models, the MoE layer usually replaces the feed-forward network (FFN) and only activates a subset of experts for each input.

While experts in MoE models are often deployed in parallel across distributed GPUs, imbalanced token-to-expert assignments lead to varying levels of expert utilization and potential latency. Despite the incorporation of balancing techniques during training, the load imbalance persists during inference. To further investigate this issue, we conduct preliminary experiments to analyze expert-specific utilization patterns and assess the impact of imbalance on practical latency.

To quantify expert utilization, we measure the load across different experts. Given an input batch $\mathbf{x} \in \mathbb{R}^{b \times s \times d}$ with batch size $b$ and sequence length $s$, the total number of tokens is $t = bs$. Since each token selects $k$ out of $n$ experts, the expected token count per expert is:

$$\bar{N} = \frac{tk}{n}. \tag{3}$$

However, due to imbalanced token assignments, some experts may receive more or fewer tokens than the expected value.

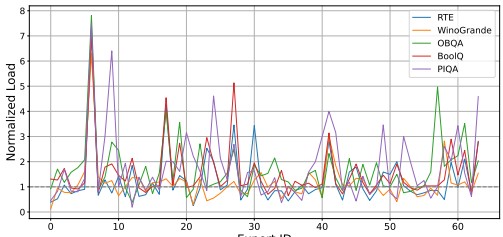

Figure 2: **Test-time expert load** of OLMoE across different datasets, where each load value is normalized by the mean load $\bar{N}$ for clarity.

Figure 2 illustrates the normalized peak token load for each expert to accommodate all tokens within a single layer of OLMoE, where some experts receive an excessively large number of tokens (e.g., more than seven times the average number of tokens), leading to severe load imbalance and, consequently, significant latency. A detailed layer-by-layer analysis is provided in Appendix F.

## 3.2 MOTIVATION – THE STRAGGLER EFFECT

Under expert parallelism, where the number of assigned tokens dictates the processing time of each expert, high-load experts become the bottleneck for overall latency within an MoE layer. Specifically, low-load experts remain idle while waiting for high-load experts to complete, leading to synchronization delays. Therefore, the latency of an MoE layer is given by:

$$L \propto max(\{N_i\}_{i=1}^n), \tag{4}$$

where $N_i$ represents the number of tokens assigned to the $i$-th expert, with the total token allocation satisfying $\sum_{i=1}^n N_i = tk$. According to Eq. 4, the latency follows the ***Straggler Effect: the most burdened experts dictate the overall latency of the MoE layer***. In the worst case, all tokens are assigned to the same group of experts, underutilizing the parallel processing capability of MoE. Conversely, distributing tokens evenly across experts maximizes computational efficiency and fully leverages the parallelism of multiple experts. With the bounds of the ideal and worst cases, the range of the highest load is given by:

$$max(\{N_i\}_{i=1}^n) \in [\bar{N}, \frac{n\bar{N}}{k}]. \tag{5}$$

However, existing MoE models often adopt a dropless strategy during inference, which fails to address token imbalance and can lead to significantly increased latency.

Given that the imbalance stems from excessively high- and low-load experts, we address this issue by exploring the following questions: (1) For ***high-load experts***, are there redundant tokens that can be dropped without causing significant performance degradation? (2) For ***low-load experts*** that must wait for high-load experts to complete forward passes, is there an opportunity to enhance their utilization and improve performance without incurring substantial additional cost?

## 4 METHODOLOGY

**Token Drop Regulates the Latency of High-Load Experts**    To address the question about overloaded experts, we first regulate their maximum utilization. Specifically, we introduce expert capacity to control token allocation. Given a capacity factor $\gamma$, the maximum number of tokens assigned to each expert (i.e., expert capacity) is defined as: $C = \gamma\bar{N}$.

A higher $\gamma$ allows more tokens to be retained, but experts handling excessive tokens may introduce latency. Conversely, a lower $\gamma$ enforces stricter capacity limits, reducing latency by discarding more tokens, but at the risk of performance degradation. With the involvement of expert capacity $\gamma$, we constrain the upper bound of latency as follows:

$$max(\{N_i\}_{i=1}^n) = \begin{cases} \gamma\bar{N} & \gamma < 1 \\ \text{within } [\bar{N}, \gamma\bar{N}] & \gamma \geq 1 \end{cases}, \tag{6}$$

where $\gamma$ is typically much smaller than $\frac{n}{k}$. This constraint ensures that no expert exceeds the specified capacity limit, effectively mitigating severe load imbalances and reducing latency. Note that tokens are distributed across devices under expert and data parallelism. To avoid additional communication overhead, we apply capacity constraints to tokens within each local device, similar to the constraints used during training (Fedus et al., 2022). This ensures that all experts respect the limits, maintaining strict control over token flow to the experts.

Specifically, when a capacity constraint is imposed on each expert, experts must evaluate the volume of assigned tokens before execution. For experts with a load below the predefined capacity, there is no difference between capacity-constrained inference and traditional inference. However, when the load exceeds the capacity, experts must discard excess tokens to adhere to the constraint. To address

this, we introduce a scoring function $\mathcal{S}$ to evaluate each token:

$$\mathcal{S}(\boldsymbol{x}) = \begin{bmatrix} s_{11} & s_{12} & \cdots & s_{1n} \\ s_{21} & s_{22} & \cdots & s_{2n} \\ \vdots & \vdots & \vdots & \vdots \\ s_{t1} & s_{t2} & \cdots & s_{tn} \end{bmatrix}, \tag{7}$$

where $s_{ij}$ denotes the importance score of the mapping from the $i$-th token to the $j$-th expert. With this score, each overflowed expert selectively discards those with lower scores. Let $\mathcal{J}$ be the set of overflowed experts and $\mathcal{S}_{\mathcal{J}}$ the corresponding columns of $\mathcal{S}$, with the Token Drop threshold set as:

$$\tau_{\mathcal{J}} = \text{KthValue}(\mathcal{S}_{\mathcal{J}}, C), \tag{8}$$

where $\tau_{\mathcal{J}}$ represents the thresholds, i.e., $C$-th highest value in $\mathcal{S}_{\mathcal{J}}$, serving as a threshold to filter out excess tokens:

$$T_{\mathcal{J}} \leftarrow \{(t, j) \mid t \in [1, \ldots, N], \ j \in \mathcal{J}, \ \mathcal{S}[t, j] \geq \tau_{\mathcal{J}}[j]\} \tag{9}$$

$$\mathcal{S}_{\mathcal{J}} \leftarrow \mathcal{S}_{\mathcal{J}} \odot M_{\mathcal{J}}, \text{ where } M_{\mathcal{J}} \leftarrow \mathbb{1}\left[\mathcal{S}_{\mathcal{J}} \geq \tau_{\mathcal{J}}\right], \tag{10}$$

where $T_{\mathcal{J}}$ denotes the token indices retained by the experts indexed in $\mathcal{J}$. The scores of rejected tokens are masked to prevent them from being routed to their corresponding overflowed experts.

Regarding the specific scoring function, we explore multiple metrics and summarize them as follows:

**Order:** Discarding later tokens once earlier tokens have filled the expert capacity. This strategy was first introduced in Switch-Transformer (Fedus et al., 2022) during training, and we extend it to the inference phase.

**Reverse Order:** Instead of discarding later tokens, this approach removes earlier tokens to comply with the expert capacity constraint.

**Random:** Dropping excess tokens randomly to meet the predefined expert capacity constraints.

**Score:** Using gating scores after the softmax and top-$k$ operations as an importance indicator and discarding tokens.

Among these metrics, "Order" and "Reverse Order" are unstable, as shuffling sequences within a batch may result in different tokens being dropped (Hayes et al., 2024). "Random" assumes all tokens have an equal probability of being dropped. In contrast, "Score" is stable, unaffected by sequence order within a batch. Notably, there is virtually no additional computational overhead associated with calculating these metrics, and the dropping operation incurs minimal cost compared to the intensive computations performed by the experts.

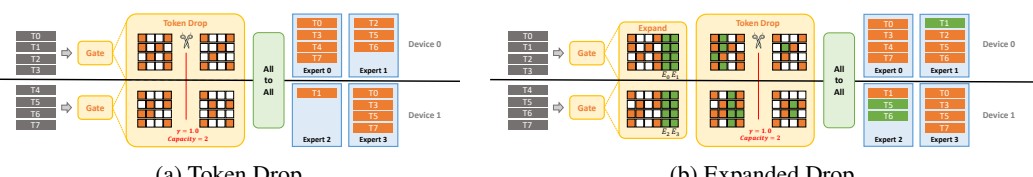

(a) Token Drop  (b) Expanded Drop

Figure 3: **Illustration of Capacity-Aware Token Drop (a) and Expanded Drop (b).** Both methods first select experts based on gating scores. In Token Drop, tokens exceeding the local device capacity are discarded prior to All-to-All communication. Expanded Drop enhances expert utilization by allowing each token to consider additional $m$ candidate experts on the same device while still enforcing strict local capacity constraints.

**Expanded Drop Enhances the Utilization of Low-load Experts**   Token Drop exclusively targets overloaded experts by discarding overflowed tokens that exceed expert capacity but does not address the underutilization of low-load experts. Next, we introduce Expanded Drop to ensure a more balanced token-to-expert allocation.

A naive approach to rerouting under-selected tokens is to mask the mapping scores of overflowed experts and then reselect experts for these tokens. However, the reselection may still result in

overflows, necessitating multiple rounds of selection and dropping, which increases latency. Moreover, the repeated selection and dropping substantially raise the cost of token-to-expert mapping.

Expanded Drop adopts a simple yet effective strategy: for each token, it selects additional candidate experts. Given $m$ experts deployed on a single GPU, a token not only selects the top-$k$ experts based on gating scores, but also includes $m$ local experts (e.g., 8 experts per device under 8-way expert parallelism across 8 GPUs for a total of 64 experts) for substitution if the initially selected experts are overflowed. As a result, each token may select up to $m + k$ experts. The final selection is then refined as experts drop tokens as needed to satisfy capacity constraints. This makes no change in the token assignments in experts that are overflowed by the top-$k$ experts. Meanwhile, for underutilized experts, the expanded top-$k + m$ candidate pool increases the likelihood of receiving additional tokens. After top-$k + m$ selection and dropping, some tokens may select more than $k$ experts. Through empirical analysis (Appendix D), we choose not to enforce a constraint that limits each token to selecting at most $k$ experts, thereby removing the need to explicitly retain the top-$k$ experts at the end.

Notably, the extra cost of token routing is minimal; the only difference lies in the negligible cost in the concatenation of the gating scores from either the top-$k$ or $m$ experts on the local device. Moreover, processing expanded tokens within the local device eliminates inter-device communication.

## 5 EXPERIMENTS

In this section, we conduct experiments under capacity-aware inference for MoE, with deployment details provided in Appendix A.

### 5.1 TOKEN DROP FOR HIGH-LOAD EXPERTS

Table 1: **Performance comparison across different capacity factors and selection metrics** (i.e., Order, Reverse Order, Random, and Score). The baseline operates without capacity constraints, represented as $+\infty$. We report the average performance over multiple random seeds.

| Method | $\gamma$ | OBQA | PIQA | RTE | WinoGrande | BoolQ | ARC-C | HellaSwag | MMLU | Avg. |
|---|---|---|---|---|---|---|---|---|---|---|
| Baseline | $+\infty$ | 45.6 | 80.1 | 53.7 | 71.2 | 74.7 | 54.5 | 79.4 | 52.5 | 64.0 |
| Order | | 42.0 | 71.5 | 53.1 | 71.2 | 74.2 | 49.5 | 76.6 | 48.4 | 60.8 |
| Reverse Order | 2.0 | 41.8 | 71.8 | 52.7 | 71.0 | 73.9 | 49.4 | 76.4 | 49.2 | 60.8 |
| Random | | 41.2 | 75.2 | 52.7 | 71.0 | 74.1 | 50.1 | 76.8 | 49.4 | 61.3 |
| Score | | **45.0** | **80.1** | **54.5** | **71.5** | **74.6** | **54.9** | **79.3** | **51.8** | **64.0** |
| Order | | 38.8 | 67.1 | 48.7 | 68.5 | 73.3 | 46.3 | 54.0 | 43.7 | 55.1 |
| Reverse Order | 1.5 | 40.2 | 67.3 | 52.7 | 70.1 | 72.7 | 45.5 | 54.4 | 45.2 | 56.0 |
| Random | | 39.6 | 72.1 | 53.8 | 68.3 | 73.8 | 45.8 | 74.2 | 45.2 | 59.1 |
| Score | | **44.8** | **77.5** | **55.2** | **70.8** | **74.3** | **53.4** | **78.6** | **50.0** | **63.1** |
| Order | | 36.0 | 60.2 | 52.2 | 62.6 | 69.6 | 38.7 | 58.0 | 36.9 | 51.8 |
| Reverse Order | 1.0 | 36.2 | 59.5 | 50.5 | 63.3 | 69.4 | 39.4 | 58.7 | 38.7 | 52.0 |
| Random | | 34.0 | 63.1 | 53.2 | 60.8 | 70.2 | 40.5 | 66.9 | 35.7 | 53.1 |
| Score | | **41.6** | **76.0** | **53.4** | **69.9** | **73.2** | **50.4** | **77.1** | **47.8** | **61.1** |

**Investigation on Token Drop Metrics**   To assess the effectiveness of different metrics in regulating token load to the target capacity, we compare various approaches on OLMoE by discarding excess tokens and applying a range of capacity factors. As shown in Table 1, varying the dropping metrics impacts performance at different levels. With higher capacities, the model maintains comparable performance even when using naive selection methods like "Random". However, as the capacity factor decreases, performance degradation becomes more pronounced, particularly for "Order", "Reverse Order", and "Random". Notably, "Score" consistently outperforms other methods by a large margin, demonstrating the effectiveness of leveraging gating scores as an importance measure. Consequently, we adopt "Score" as the default metric.

**Efficiency Gains from Capacity-Constrained Inference**   We next explore the efficiency improvements achieved by imposing expert capacity. Specifically, we employ distributed inference using eight H20 GPUs, utilizing an 8-way Data Parallelism (DP) and 8-way Expert Parallelism (EP) strategy

Figure 4: Speedup of a single MoE layer compared to the baseline without capacity constraints, achieved through two capacity-aware inference methods: Token Drop and Expanded Drop.

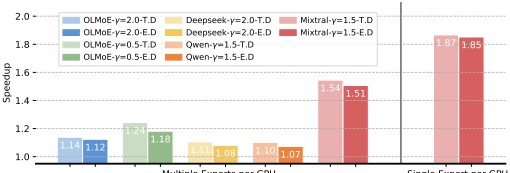

Figure 5: End-to-end speedup. "T.D." and "E.D." are abbreviations for Token Drop and Expanded Drop, respectively.

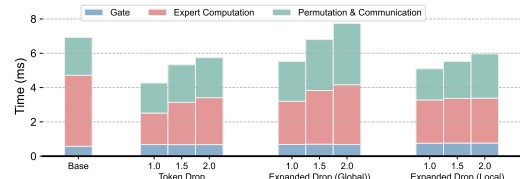

Figure 6: Breakdown analysis of the inference latency on OLMoE with different capacity factors (e.g., 1.0, 1.5, 2.0).

through the Megatron-LM framework (Shoeybi et al., 2019). The input batches are configured with a batch size of 8K and a sequence length of 512, simulating real-time serving scenarios with high query throughput. Notably, in Mixtral-8×7B-Instruct model, each GPU typically hosts one or two experts, whereas, in models like OLMoE-Instruct, multiple experts must be deployed on a single GPU (e.g., eight experts per GPU) due to GPU resource constraints.

As illustrated in Figure 4, imposing constraints on expert capacity through Token Drop and Expanded Drop, considerably accelerates inference across the four tested MoE models, in comparison to the baseline model without capacity limitations. The enhanced efficiency of each MoE layer (Figure 4) contributes to faster end-to-end model inference (Figure 5). Moreover, as the capacity factor $\gamma$ decreases, capacity-aware inference methods achieve significantly greater acceleration.

Notably, the efficacy of acceleration is influenced by the numerical relationship between the total experts and the engaged GPUs in Expert Parallelism. As illustrated in Figure 5, for Mixtral-8×7B-Instruct, deploying one or two experts per GPU maximizes the effectiveness of capacity-aware inference. In this configuration, Token Drop and Expanded Drop achieve end-to-end model speedups of 1.87 × and 1.85×, respectively, with $\gamma = 1.5$. Conversely, deploying a greater number of experts on a single GPU results in more modest acceleration gains, as evidenced by the "8E/GPU" (OLMoE-Instruct and DeepSeek-V2-Lite) and "10E/GPU" (Qwen1.5-MoE-Chat) settings in Figure 4 and Figure 5. This is because the aggregated load from multiple experts diminishes the proportion of reduced load, which is achieved by limiting the straggler expert. Therefore, it is anticipated that allocating more GPUs for expert distribution, thereby reducing the number of experts per GPU, would enhance the acceleration effect of capacity-aware inference.

The breakdown analysis presented in Figure 6 demonstrates that our proposed capacity-aware inference methods substantially reduce the duration of expert computation, permutation, and communication, while preserving a comparable cost for gate processing. Notably, the durations of permutation and communication increase when tokens are expanded across a range of global experts. This is due to the increased communication workload required to transmit expanded global tokens across various GPU devices. Consequently, these results underscore the necessity of restricting the expanded tokens to be processed by local experts.

**Mitigating the Straggler Effect with Minimal Token Discarding** Given that expert capacity enforces MoE layers to discard overflowed tokens, we next establish the relationship between expert capacity and the corresponding number of dropped tokens. For a capacity factor $\gamma$, the total proportion

of dropped tokens is given by:

$$DT = \frac{\sum_{i=1}^{n} \text{ReLU}(N_i - \gamma \bar{N})}{\sum_{i=1}^{n} N_i},\tag{11}$$

where $\text{ReLU}(N_i - \gamma \bar{N})$ represents the number of dropped tokens for the $i$-th expert.

Figure 7 visualizes the number of dropped tokens across different capacity factors for various test datasets, with a more detailed illustration provided in Appendix G. Although the most overloaded expert receives much more tokens than the expected number of tokens $\bar{N}$, regulating the maximum capacity has a limited impact on the overall number of accommodated tokens, thereby maintaining competitive performance even after discarding overflow tokens. Moreover, dropping a small proportion of overflowed tokens can significantly reduce the latency caused by overloaded experts (e.g., dropping 12% of overloaded tokens improves the inference speed by 85% in Mixtral-8×7B-Instruct), highlighting the efficacy of capacity-aware inference in improving both performance and efficiency.

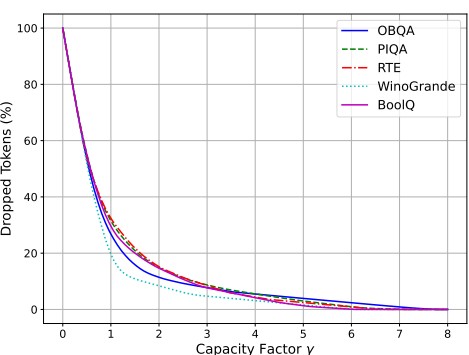

Figure 7: Analysis of dropped tokens with respect to capacity factors.

## 5.2 EXPANDED DROP TO LOW-LOAD EXPERTS

Some experts receive very few tokens, raising the question of whether they are redundant or can be leveraged for balanced allocation. We next examine their role and validate the effectiveness of Expanded Drop.

Table 2: **Comparison of Expert Drop, Token Drop, and Expanded Drop**. The capacity factor $\gamma$ is set to 2.0 for OLMoE and DeepSeek-V2-Lite, and 1.5 for Qwen1.5-MoE-Chat and Mixtral-8×7B-Instruct. For Expert Drop, each forward pass skips one out of eight experts for Mixtral-8×7B-Instruct, and the bottom 10% of lowest-load experts for other models.

| Model | Method | OBQA | PIQA | RTE | WinoGrande | BoolQ | ARC-C | HellaSwag | MMLU | GSM8K | Avg. |
|---|---|---|---|---|---|---|---|---|---|---|---|
| | Baseline | 47.6 | 80.2 | 67.9 | 69.9 | 80.7 | 57.0 | 80.6 | 52.8 | 35.1 | 63.5 |
| OLMoE-Instruct | Expert Drop | 44.6 | 76.9 | 64.0 | 67.6 | 78.2 | 54.4 | 77.0 | 50.6 | 31.6 | 60.5 |
| | Token Drop | **47.8** | 77.9 | 64.6 | 69.2 | 80.0 | **57.2** | 79.7 | 51.5 | 32.4 | 62.3 |
| | Expanded Drop | 47.2 | **79.4** | **66.3** | **70.5** | **80.9** | 57.1 | **80.3** | **52.3** | **34.4** | **63.2** |
| | Baseline | 42.4 | 79.9 | 72.9 | 70.0 | 81.3 | 54.1 | 80.4 | 59.8 | 52.0 | 65.9 |
| Qwen1.5-MoE-Chat | Expert Drop | 41.4 | 78.7 | 71.2 | 68.6 | 80.6 | 52.9 | 79.1 | 58.1 | 49.4 | 64.4 |
| | Token Drop | 40.4 | 78.8 | **72.6** | 69.1 | 80.9 | 53.0 | 80.0 | **59.3** | 51.9 | 65.1 |
| | Expanded Drop | **43.4** | **79.1** | **72.6** | 69.6 | **81.1** | **53.4** | **80.3** | 59.3 | **52.1** | **65.6** |
| | Baseline | 45.4 | 81.4 | 72.6 | 75.5 | 82.9 | 61.0 | 81.5 | 57.3 | 66.4 | 69.3 |
| DeepSeek-V2-Lite-Chat | Expert Drop | 41.8 | 77.6 | 71.9 | 72.5 | 81.6 | 57.1 | 75.5 | 53.3 | 56.0 | 65.3 |
| | Token Drop | 45.2 | 78.3 | 72.6 | 74.0 | 83.2 | 59.3 | 80.9 | **57.3** | 62.7 | 68.2 |
| | Expanded Drop | **45.4** | **79.4** | **73.3** | **75.4** | **83.2** | **60.4** | **81.5** | 57.2 | **64.1** | **68.9** |
| | Baseline | 47.4 | 84.8 | 71.8 | 82.5 | 88.5 | 71.7 | 87.5 | 70.2 | 64.2 | 74.3 |
| Mixtral-8×7B-Instruct | Expert Drop | 46.8 | 83.2 | 70.1 | 81.3 | 87.6 | 67.1 | 85.6 | 66.2 | 62.3 | 72.2 |
| | Token Drop | 46.4 | 83.3 | 71.7 | 82.2 | 88.3 | 71.2 | 87.4 | 69.1 | 64.7 | 73.8 |
| | Expanded Drop | **47.8** | **85.0** | **71.8** | **83.0** | **88.6** | **71.5** | **87.6** | **70.2** | **64.6** | **74.5** |

**The Critical Role of Low-Load Experts** To explore the impact of low-load experts, we further compare dropping tokens (i.e., Token Drop) with skipping experts (i.e., Expert Drop). For Expert Drop, we adopt a conservative strategy that dynamically skips the 10% of experts with the lowest token loads. Notably, the proportion of tokens removed in Expert Drop is significantly lower than in Token Drop (2% in Expert Drop vs. 12% in Token Drop on OLMoE-Instruct).

Despite this, as shown in Table 2, Expert Drop experiences significant performance degradation and is outperformed by Token Drop by a large margin. Moreover, due to the small proportion of tokens assigned to low-load experts, removing these experts provides only marginal improvements in

inference speed (less than a 5% speedup). These findings indicate that retaining low-load experts better preserves the performance of MoE models.

To analyze expert selection and justify Expanded Drop, we sort, for each token, all experts by their gating scores in descending order and record the ranked scores (top-1, 2, ..., top-$N$). Aggregating across tokens, we compute the average, maximum, and minimum score at each rank (Figure 8). The curves show that while top-ranked experts receive much higher scores, the decay across ranks is gradual rather than abrupt, yielding a relatively flat tail beyond the first few ranks. Consequently, experts just outside the top-$k$ often have scores comparable to those near the bottom of the top-$k$, enabling rerouting or dropping without materially changing model behavior,

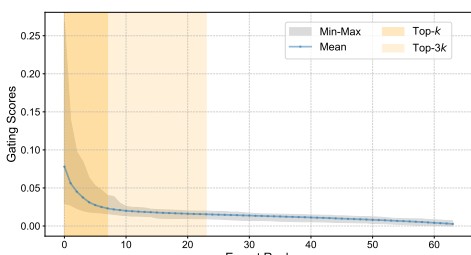

Figure 8: Gating score distribution across ranked experts.

i.e., rerouted tokens still go to reasonably relevant experts with similar gating probabilities. This validates the design of Expanded Drop, as it can exploit the flatness of the gating distribution to balance load without sacrificing accuracy.

**Effectiveness of Expanded Drop** We examine the effectiveness of utilizing low-load experts by Expanded Drop instead of simply discarding these tokens to meet the target capacity. Comparing Expanded Drop with Token Drop, redistributing excess tokens to low-load experts enhances performance, yielding a 0.9% improvement in the average performance of Qwen1.5-MoE-Chat. Furthermore, considering the performance degradation observed in Expert Drop, our findings highlight the crucial role of low-load experts in maintaining model effectiveness.

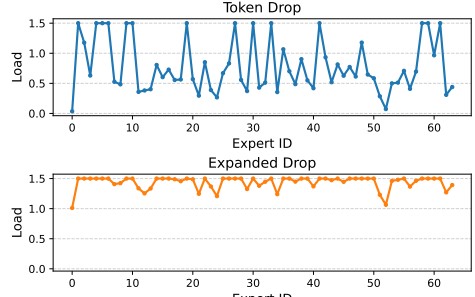

Figure 9: Normalized expert load after Token Drop and Expanded Drop.

Expanded Drop overselects experts for each token to expand the selection scope and encourage more balanced token-expert assignments. As shown in Figure 9, increasing the overselection ratio $m$ allows tokens to consider more candidate experts after being dropped from overflowed ones, thereby improving low-load expert utilization and balancing the expert load.

**Advanced Variant: Device-Level Capacity-Aware Inference** In scenarios where multiple experts reside on the same device, the overall straggler effect is determined by the *aggregated* load on each device. Expert-level capacity constraints impose a strict limit. Specifically, the number of tokens assigned to a single expert must not exceed $\gamma \bar{N}$. When extended to a device with $n_l$ local experts, this implies that the total load is bounded by $n_l \cdot \gamma \bar{N}$. However, this strict constraint can lead to over-aggressive token dropping, as a single overloaded expert may force token removal even when sufficient spare capacity remains across other experts on the same device.

To alleviate this issue, we introduce a **device-level capacity-aware** formulation, which applies the constraint at the device granularity rather than on individual experts. For instance, we enforce $N_1 + N_2 + \cdots + N_{n_l} \leq n_l \cdot \gamma \bar{N}$. Unlike expert-level constraints, which may drop tokens simply because a single expert exceeds its local budget, the device-level constraint admits these tokens as long as the *total* device-level load remains within the allowable bound. This results in a smoother, less restrictive, and more utilization-efficient alternative.

Table 3: Comparison of Device-Level and Expert-Level capacity-aware inference on Qwen3-MoE.

| Method | Granularity | $\gamma$ | OBQA | PIQA | RTE | WinoGrande | BoolQ | ARC-C | HellaSwag | MMLU | GSM8K | Avg. |
|---|---|---|---|---|---|---|---|---|---|---|---|---|
| Baseline | – | $+\infty$ | 45.2 | 80.6 | 82.3 | 69.5 | 88.6 | 69.5 | 77.6 | 77.9 | 89.5 | 75.6 |
| Expanded Drop | Expert | 1.5 | **43.6** | 79.4 | 80.3 | 68.4 | 86.5 | 68.8 | 74.4 | 76.8 | 87.0 | 73.9 |
| | Device | 1.0 | 42.4 | **79.7** | **81.2** | **69.3** | **88.0** | **69.4** | **77.1** | **77.1** | **88.9** | **74.8** |

We evaluate this variant and observe that device-level constraints consistently outperform expert-level constraints in downstream performance, as shown in Table 3. This more flexible constraint also leads to stronger practical speedups: on Qwen3-MoE (Yang et al., 2025) with $\gamma = 1$, the device-level formulation achieves a $1.31\times$ end-to-end speedup and a $1.51\times$ speedup on a single MoE layer, both surpassing the $1.23\times$ and $1.40\times$ speedups obtained under expert-level constraints with $\gamma = 1.5$.

## 5.3 EXTENSION TO MULTIMODAL MIXTURE OF EXPERTS

In addition to applying capacity-aware inference to MoE models for language tasks, we also explore its effectiveness in multimodal MoE settings. Specifically, we evaluate the OLMoE-based MolmoE (Deitke et al., 2024) across multimodal benchmarks, including MME (Fu et al., 2023), MMBench (Liu et al., 2023), and SEED-Bench (Li et al., 2024a).

Given that the input sequence contains tokens from multiple modalities, we first investigate different token dropping strategies. Specifically, we first treat all tokens equally and drop those with the lowest scores ("Uniform"). Beyond this, considering the redundancy often found in image tokens, we also experiment with a strategy that prioritizes dropping image tokens before selectively removing text tokens ("Image First"). For comparison, we also consider dropping text tokens first ("Text First"). As shown in Table 4, on the MME benchmark, this image-first strategy yields improved performance, highlighting the benefit of prioritizing image-token dropping for load balance in multimodal MoE models.

Table 4: **Capacity-aware inference for multimodal MoE models**. "Percep." and "Cognit." denote Perception and Cognition, respectively. $\gamma$ is set to 1.0.

| Method | Strategy | Percep. | Cognit. |
|---|---|---|---|
| Baseline | – | 1358.1 | 269.6 |
| Token Drop | Uniform | 1248.4 | 245.4 |
| Expanded Drop | | 1307.6 | 273.6 |
| Token Drop | Text First | 1114.2 | 214.4 |
| Expanded Drop | | 1163.6 | 241.3 |
| Token Drop | Image First | 1346.5 | 288.9 |
| Expanded Drop | | 1362.1 | 297.1 |

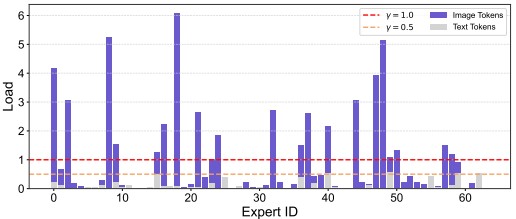

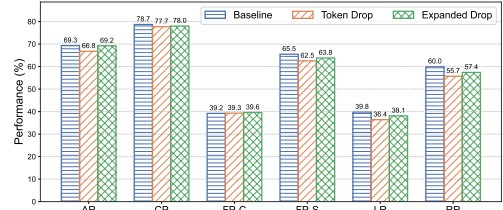

Figure 10: Multimodal token assignments across different experts.

Figure 11: Comparison on MMBench across six multimodal capabilities in Appendix C.

Given the redundancy of image tokens and their large proportion in multimodal tasks, we further investigate more aggressive capacity factors for Token Drop and Expanded Drop using the "Image First" strategy. Figure 11 demonstrates the effectiveness of Capacity-Aware Inference under low capacity constraints (i.e., $\gamma = 0.5$). This is largely due to the high redundancy in image tokens (Chen et al., 2024), which allows a higher dropping ratio without significantly affecting performance. The dominance of image tokens also enables the use of very low capacity factors without significantly affecting text token retention, as illustrated in Figure 10. Dropping image tokens at higher ratios leads to more balanced token assignments and substantially improved inference efficiency.

## 6 CONCLUSION

In this paper, we identify the issue of imbalanced token-to-expert assignment in Mixture of Experts (MoE) models and introduce the Straggler Effect during inference, where high-load experts become efficiency bottlenecks and dictate overall latency. To address this problem, we propose Capacity-Aware Token Drop, which mitigates expert overload by enforcing strict capacity constraints. Additionally, to better utilize underloaded experts, we present Capacity-Aware Expanded Drop, which allows tokens to select additional experts on the same device while still respecting capacity limits, thereby improving expert utilization. Our findings and proposed methods offer valuable insights and effective strategies for improving MoE inference efficiency.

## 7 ETHICS STATEMENT

Our work focuses on improving the efficiency of large-scale Mixture-of-Experts (MoE) models at inference time. The proposed methods are lightweight modifications to model execution and do not involve collecting or annotating new data. All experiments are conducted on publicly available pretrained models and standard open-source benchmarks. We emphasize that our contributions are intended solely for research and educational purposes in efficient machine learning. We will explicitly release code under a research license and include clear terms discouraging misuse in sensitive applications such as mass surveillance or privacy-intrusive deployments.

## 8 REPRODUCIBILITY STATEMENT

We are committed to ensuring full reproducibility of our results. Upon acceptance, we will release the complete implementation of Capacity-Aware Inference, including inference-time strategies (Token Drop and Expanded Drop), evaluation harness integration, and scripts for running all benchmarks. Detailed instructions are provided in a reproducibility README covering environment setup, library versions, random seeds, and commands for end-to-end replication.

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

## A IMPLEMENTATION DETAILS

**Models** We mainly focus on lightweight MoE models (less than 20B parameter budget). We conduct experiments on OLMoE (Muennighoff et al., 2024), Qwen1.5-MoE (Team, 2024b), DeepSeek-V2-Lite (et al., 2024a), Mixtral (Jiang et al., 2024), MolmoE (Deitke et al., 2024), and Qwen3-MoE (Yang et al., 2025), due to their competitive performance and widespread applications.

**Datasets** To evaluate model performance, we report normalized zero-shot or few-shot accuracy on the LM-Harness benchmark. The number of shots for each task is detailed in Table 5, which includes multiple tasks: ARC-C (Clark et al., 2018), BoolQ (Clark et al., 2019), HellaSwag (Zellers et al., 2019), MMLU (Hendrycks et al., 2021), OBQA (Mihaylov et al., 2018a), PIQA (Bisk et al., 2019), RTE (Wang et al., 2019), Wino-Grande (ai2, 2019), and GSM8K (Cobbe et al., 2021). The evaluation code is based on EleutherAI's LM Harness framework (Gao et al., 2023).

Table 5: **Experimental settings for evaluated language tasks.** "Norm" refers to the normalization performed with respect to the length of the input.

| Task | Number of few-shot | Metric |
|---|---|---|
| BoolQ | 0 | Accuracy |
| RTE | 0 | Accuracy |
| OBQA | 0 | Accuracy (Norm) |
| PIQA | 0 | Accuracy (Norm) |
| MMLU | 5 | Accuracy |
| WinoGrande | 5 | Accuracy |
| GSM8K | 5 | Exact Match |
| HellaSwag | 10 | Accuracy (Norm) |
| ARC-C | 25 | Accuracy (Norm) |

## B ADDITIONAL RESULTS ON LANGUAGE TASKS

Long-context understanding poses substantial challenges for language models. We therefore investigate whether Token Drop and Expanded Drop can maintain competitive performance in this setting. Table 6 presents their results on a long in-context learning benchmark. Following LongICLBench (Li et al., 2024b), we evaluate on the BANKING77 dataset (Casanueva et al., 2020), with the only modification being that we sample 100 test examples. BANKING77 is an intent detection dataset in the banking domain with 77 classes. We consider 1-shot/label to 5-shot/label settings, corresponding to context lengths of 2K, 4K, 7K, 9K, and 14K tokens. Across different input sequence lengths, both Token Drop and Expanded Drop consistently match the performance of the baseline. This demonstrates the strong generalization ability and wide applicability of these techniques.

Table 6: Performance of Token Drop and Expanded Drop on DeepSeek-V2-Lite in LongICLBench.

| Method | 2K | 4K | 7K | 9K | 11K | Avg. |
|---|---|---|---|---|---|---|
| Baseline | 0.28 | 0.44 | 0.56 | 0.62 | 0.70 | 0.52 |
| Token Drop | 0.27 | 0.43 | 0.57 | 0.65 | 0.71 | 0.53 |
| Expanded Drop | 0.23 | 0.53 | 0.63 | 0.62 | 0.70 | 0.54 |

To further assess the robustness of Token Drop and Expanded Drop in generation tasks, we test additional benchmarks such as HumanEval (Chen et al., 2021), NQ-Open (Kwiatkowski et al., 2019), and MTS-Dialog (Ben Abacha et al., 2023) in Table 7.

Table 7: Performance of Token Drop and Expanded Drop across three MoE-based LLMs on HumanEval (HE), NQ-Open (NQ), and MTS-Dialog (MTS). The capacity factor is set as 1.0 here and we report pass@1 for HE, exact_match for NQ and BERTScore for MTS, respectively.

| Method | OLMoE-Instruct | | | Qwen1.5-MoE-Chat | | | DeepSeek-V2-Lite-Chat | | |
|---|---|---|---|---|---|---|---|---|---|
| | HE | NQ | MTS | HE | NQ | MTS | HE | NQ | MTS |
| Baseline | 0.311 | 0.175 | 0.827 | 0.280 | 0.226 | 0.874 | 0.506 | 0.265 | 0.830 |
| Token Drop | 0.299 | 0.159 | 0.823 | 0.281 | 0.214 | 0.869 | 0.494 | 0.252 | 0.831 |
| Expanded Drop | 0.303 | 0.170 | 0.829 | 0.291 | 0.224 | 0.872 | 0.508 | 0.268 | 0.831 |

The consistent performance on code generation, open-domain question answering, and dialog creation demonstrates the robustness of Token Drop and Expanded Drop for long-range generation tasks.

# C   ADDITIONAL RESULTS ON MULTIMODAL TASKS

In the scope of this paper, multimodal tasks refer to those involving both vision and language modalities. We evaluate model performance using three representative benchmarks: MME, MMBench, and SEED-Bench, each targeting different aspects of multimodal understanding and reasoning.

MME benchmark evaluates vision-language models along two dimensions: perception, which tests visual grounding and recognition, and cognition, which assesses reasoning abilities such as counting and relational understanding. It provides a fine-grained analysis of multimodal understanding.

MMBench is a comprehensive benchmark designed to assess the general multimodal understanding ability of vision-language models. It evaluates model performance across six core capabilities: Coarse Perception (CP), Fine-grained Perception–including single-instance (FP-S) and cross-instance (FP-C), Attribute Reasoning (AR), Logical Reasoning (LR), and Relational Reasoning (RR). By covering both perception and reasoning-oriented tasks, MMBench provides detailed insights into the strengths and limitations of VLMs across diverse multimodal scenarios.

SEED-Bench is a large-scale benchmark for evaluating the generative comprehension of Multimodal Large Language Models (MLLMs) across both image and video modalities. It includes 19K human-annotated multiple-choice questions spanning 12 evaluation dimensions, enabling objective and efficient assessment without human or GPT intervention. SEED-Bench reveals model limitations and maintains a public leaderboard to support fair comparison and future research.

| Component | Content | Token Count |
|---|---|---|
| Image |  | 576 |
| Text Prompt | `Is this artwork titled virgin and child with sts catherine, cecilia, barbara, and ursula? Please answer yes or no.` | 31 |
| **Total** | – | **607** |

Table 8: An example multimodal query in the MME benchmark, showing the dominant proportion of image tokens compared to text tokens.

As shown in Table 8, these tasks typically introduce a large number of image tokens. When faced with imbalanced token-to-expert assignments, dropping redundant image tokens significantly improves load balancing. Moreover, due to the high redundancy among image tokens, dropping a portion of them has minimal impact on model performance.

As in MME and MMBench, the Image-First variants of Token Drop and Expanded Drop also exhibit consistent effectiveness on SEED-Bench (Li et al., 2024a), maintaining strong performance even under low capacity factors such as $\gamma = 0.5$. In addition to the redundancy in image tokens, Figure 10 shows that text tokens constitute only a small portion of the total token assignments. This allows the regulation of $\gamma$ to retain almost all text tokens under the Image-First strategy.

| Method | $\gamma$ | Inst. Attr. | Inst. ID | Inst. Interact. | Inst. Loc. | Inst. Count | Scene | Spatial | Text | Reasoning | Overall |
|---|---|---|---|---|---|---|---|---|---|---|---|
| Baseline | $\infty$ | 74.2 | 71.4 | 58.8 | 62.8 | 57.0 | 73.5 | 49.6 | 72.6 | 76.4 | 68.7 |
| Token Drop | 0.5 | 70.4 | 67.8 | **60.8** | 57.8 | 51.8 | **71.5** | 43.5 | 58.3 | 71.3 | 64.9 |
| Expanded Drop | | **71.2** | 67.3 | 57.7 | **58.8** | **53.6** | 71.0 | **45.2** | **64.3** | **71.9** | **65.5** |
| Token Drop | 1.0 | 73.6 | 70.2 | 58.8 | 62.6 | **56.9** | 72.7 | 48.1 | 61.9 | 73.4 | 68.0 |
| Expanded Drop | | **73.8** | **70.7** | **59.8** | **63.5** | 56.7 | **73.1** | **49.6** | **70.2** | **74.6** | **68.4** |
| Token Drop | 1.5 | 73.5 | **71.2** | 58.8 | 62.9 | 57.1 | 73.1 | 48.7 | 65.5 | 74.6 | 68.3 |
| Expanded Drop | | **73.7** | 71.0 | **61.9** | **64.7** | **57.4** | **73.3** | **49.3** | **71.4** | **76.1** | **68.7** |

Table 9: Token Drop and Expanded Drop strategies for multimodal MoE models evaluated on SEED-Bench. Abbreviations: Inst. Attr. = Instance Attributes; Inst. ID = Instance Identification; Inst. Interact. = Instance Interaction; Inst. Loc. = Instance Localization; Inst. Count = Instance Counting.

# D    ABLATION STUDY

**Model-Specific Imbalanced Property**    We explore the imbalance property in various models, such as OLMoE, DeepSeek, Qwen, and Mixtral, which differ in both architectures (e.g., depth and width) and training strategies (e.g., training from scratch (Muennighoff et al., 2024; et al., 2024a) vs. training after upcycling (Jiang et al., 2024; Team, 2024b)).

On the one hand, our findings in Appendix F reveal that different training strategies result in significantly varying levels of imbalance. Specifically, MoE models trained from scratch exhibit a much higher degree of imbalance. For instance, OLMoE and DeepSeek-V2-Lite experience peak expert-wise token allocations exceeding $5\bar{N}$, whereas Qwen1.5-MoE and Mixtral are upcycled from dense language models and maintain a more balanced distribution, with peak expert-wise allocations staying below $3\bar{N}$. This is because upcycling initializes all experts with identical parameters (Komatsuzaki et al., 2023), reducing divergence and promoting balanced training in the early stages.

On the other hand, despite the widespread use of auxiliary balance loss in MoE training, it does not guarantee balanced token assignments across experts, as token distribution still varies significantly during inference on test data. This necessitates integrating expert capacity into the inference process.

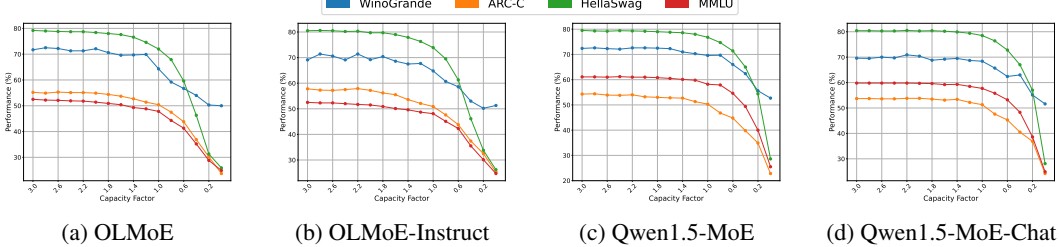

| (a) OLMoE | (b) OLMoE-Instruct | (c) Qwen1.5-MoE | (d) Qwen1.5-MoE-Chat |

Figure 12: Performance change as capacity factors decrease from 3.0 to 0.0.

**Capacity Factor**    Beyond the specific capacity values presented in Table 1, we further investigate a wide range of capacity factors in Figure 12, spanning from 0.0 to 3.0. We exclude values exceeding 3.0, as their performance closely aligns with capacity-agnostic scenarios. By analyzing the performance changes when decreasing the capacity factor, we find that setting $\gamma$ to 1.5 is sufficient to maintain performance comparable to the original models. However, maintaining performance becomes challenging under excessively low capacity factors, as high-load experts are forced to drop a significant number of tokens.

**Speedup Measurement Under Different Workloads**    To assess the effectiveness of Token Drop in efficiency, we measure speedup under a range of token workloads. Specifically, we vary both the sequence length and batch size, and present the speedup results on DeepSeek-V2 with a capacity factor of 2.0 in Table 10.

| Batch Size | 8K | 8K | 8K | 4K | 2K | 2K | 1K | 1K | 1K |
| Prompt Length | 0.1K | 0.2K | 0.4K | 1K | 1K | 2K | 1K | 2K | 4K |
|---|---|---|---|---|---|---|---|---|---|
| Speedup | 1.09× | 1.18× | 1.24× | 1.26× | 1.27× | 1.27× | 1.27× | 1.24× | 1.23× |

Table 10: Speedup results across varying batch sizes and prompt lengths.

The straggler effect becomes more pronounced under heavier workloads, where GPUs operate at higher utilization with limited spare capacity, making the speedup more noticeable. In practical server-side MoE deployments, workloads are substantially higher, and our techniques effectively mitigate the resulting straggler effect.

**Maximum Number of Selected Experts**    Expanded Drop adopts an overselection-and-dropping strategy that not only maintains load balance but also improves the utilization of underloaded experts. Although this mechanism allows some tokens to select more than $k$ experts, Table 11 demonstrates that such flexibility benefits downstream task performance, suggesting that enforcing a strict maximum of $k$ experts is unnecessary. Allowing additional expert selections can enhance representational capacity, whereas rigid constraints may unnecessarily limit model performance.

Table 11: **Ablation study on limiting the maximum number of** $k$ **selected experts.** "w/max" and "w/o max" indicate runs *with* and *without* this constraint, respectively. $\gamma$ is set to 1.0.

| $m$ | Method | OBQA | PIQA | RTE | WinoGrande | BoolQ | ARC-C | HellaSwag | MMLU | Avg. |
|---|---|---|---|---|---|---|---|---|---|---|
| $2k$ | w/ max | 42.4 | 75.8 | 53.2 | 68.6 | 72.7 | 50.3 | 77.1 | 47.6 | 61.0 |
| | w/o max | 42.2 | 75.9 | 53.4 | 69.7 | 73.1 | 50.3 | 77.0 | 47.8 | 61.2 |
| $3k$ | w/ max | 42.0 | 75.6 | 53.4 | 69.6 | 72.7 | 50.3 | 77.0 | 47.4 | 61.0 |
| | w/o max | 42.0 | 75.6 | 53.8 | 69.8 | 72.9 | 50.3 | 77.1 | 47.6 | 61.1 |

## E   IMPLEMENTATION-LEVEL CODE FOR TOKEN DROP AND EXPANDED DROP

The detailed implementation-level code for Token Drop and Expanded Drop is presented in Algorithms 1 and 2, respectively.

**Algorithm 1** Token Drop

```
# x: input tokens;
# k: top-k experts;
# gamma: capacity factor;

def token_drop(self, x, k, gamma):
  logits = self.gate(x)
  scores = torch.softmax(logits, dim=-1)

  topk_scores, topk_idx = scores.topk(k, dim=1)

  topk_mask = torch.zeros_like(scores).scatter(1,
      topk_idx, 1)
  masked_scores = scores * topk_mask

  N, E = scores.size()
  cap = int(gamma * (N * k) / E)

  _, keep_idx = masked_scores.topk(cap, dim=0)
  cap_mask = torch.zeros_like(scores).scatter(0,
      keep_idx, 1)

  final_map = topk_mask * cap_mask
  final_scores = masked_scores * final_map

  return final_scores, final_map
```

**Algorithm 2** Expanded Drop

```
# x: input tokens;
# k: top-k experts;
# gamma: capacity factor;
# local_ids: local device expert ids;

def expanded_drop(self, x, k, gamma, local_ids):
  logits = self.gate(x)
  scores = torch.softmax(logits, dim=-1)

  topk_scores, topk_idx = scores.topk(k, dim=1)

  N, E = scores.size()
  local_idx = torch.tensor(local_ids, device=x.
      device).repeat(N, 1)
  exp_idx = torch.cat([topk_idx, local_idx], dim
      =1)

  local_scores = scores[:, local_ids]
  exp_scores = torch.cat([topk_scores,
      local_scores], dim=1)

  exp_mask = torch.zeros_like(scores).scatter(1,
      exp_idx, 1)
  masked_scores = scores * exp_mask

  cap = int(gamma * (N * k) / E)

  _, keep_idx = masked_scores.topk(cap, dim=0)
  cap_mask = torch.zeros_like(scores).scatter(0,
      keep_idx, 1)

  final_map = exp_mask * cap_mask
  final_scores = masked_scores * final_map

  return final_scores, final_map
```

## F   LAYER-WISE EXPERT LOAD

To analyze imbalanced token assignments, we measure the expert load for each expert by tracking the peak expert load while running MoE models on various test datasets. Figure 13, 14, 15 and 16 present the full results for the normalized layer-wise expert load for OLMoE, DeepSeek-V2, Qwen1.5-MoE, and Mixtral-8×7B-Instruct, respectively.

## G   CALCULATION OF DROPPED TOKENS

Based on Equation 11, we calculate the total number of dropped tokens across experts in each layer under different capacity factors, as illustrated in Figures 17, 19, 18, and 20.

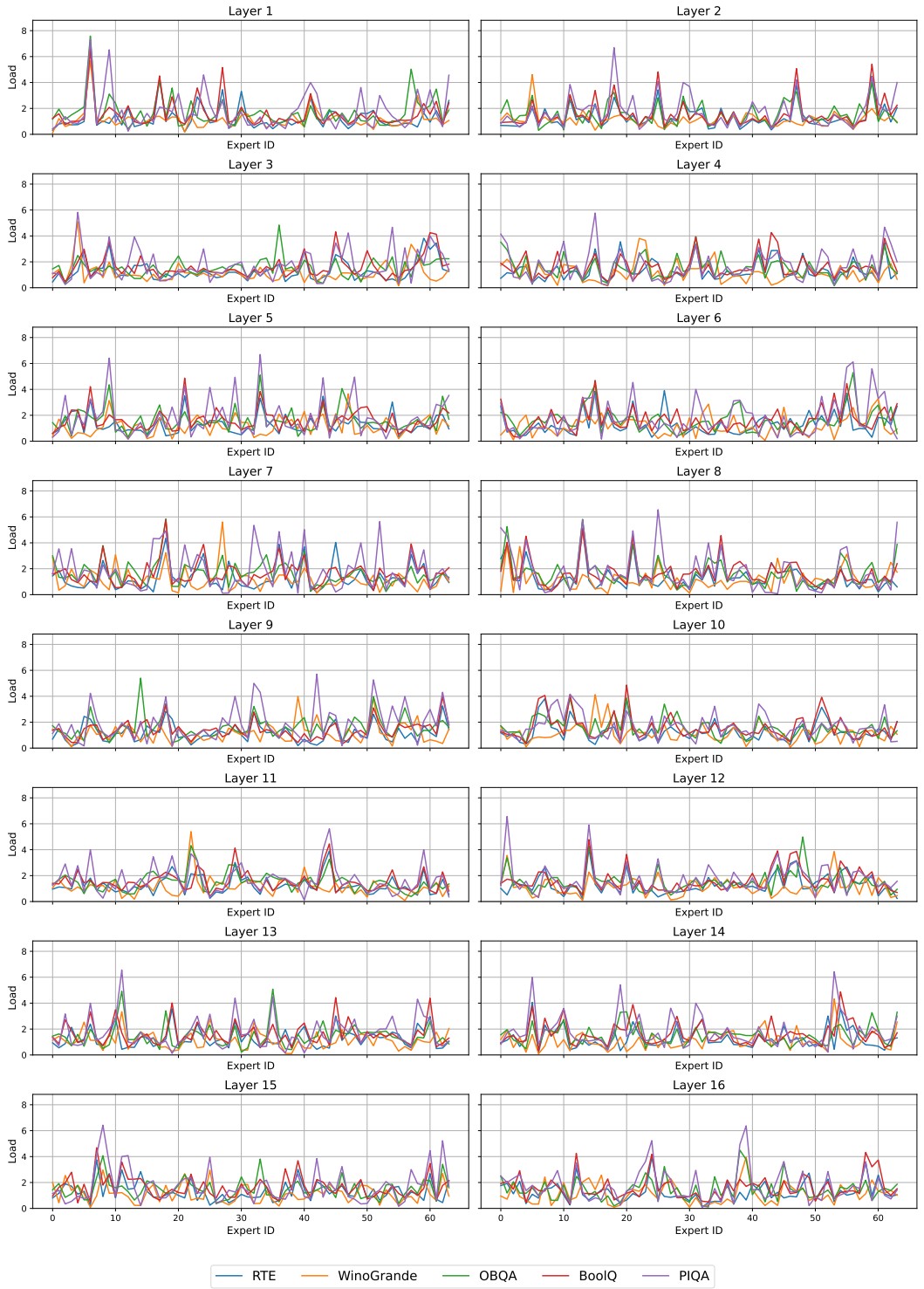

Figure 13: Layer-wise expert load in OLMoE-Instruct.

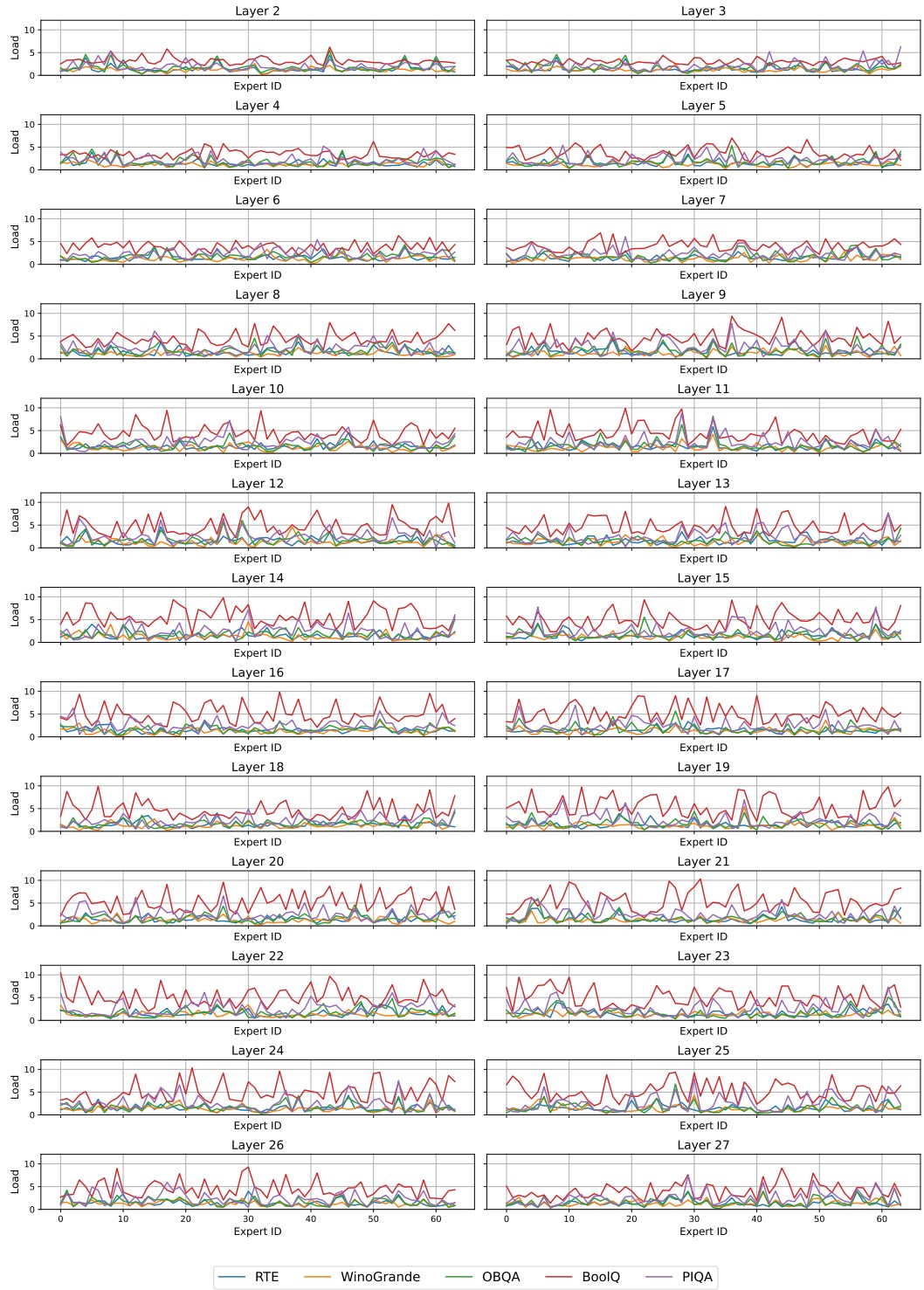

Figure 14: Layer-wise expert load in DeepSeek-V2-Lite.

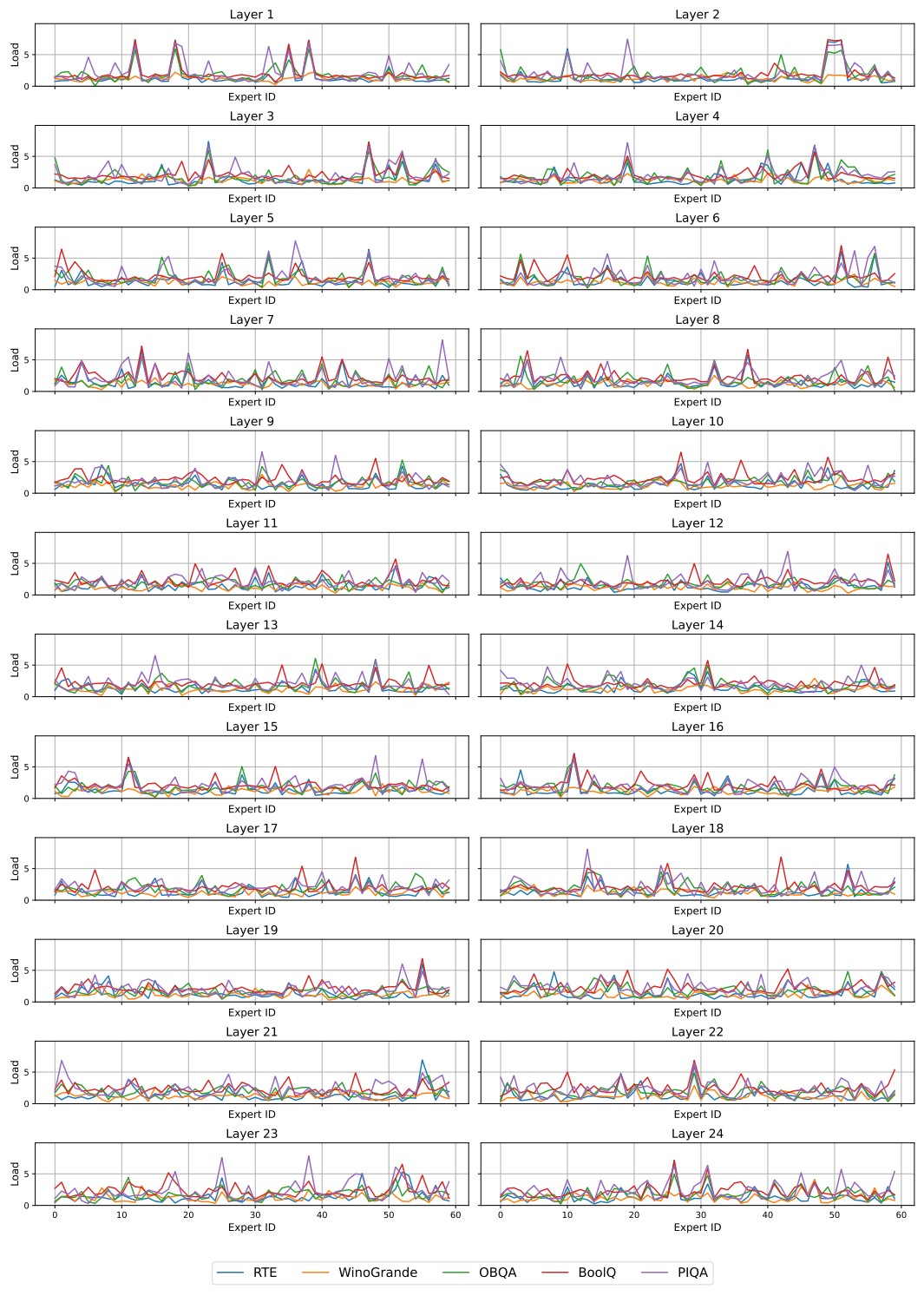

Figure 15: Layer-wise expert load in Qwen1.5-MoE-Chat.

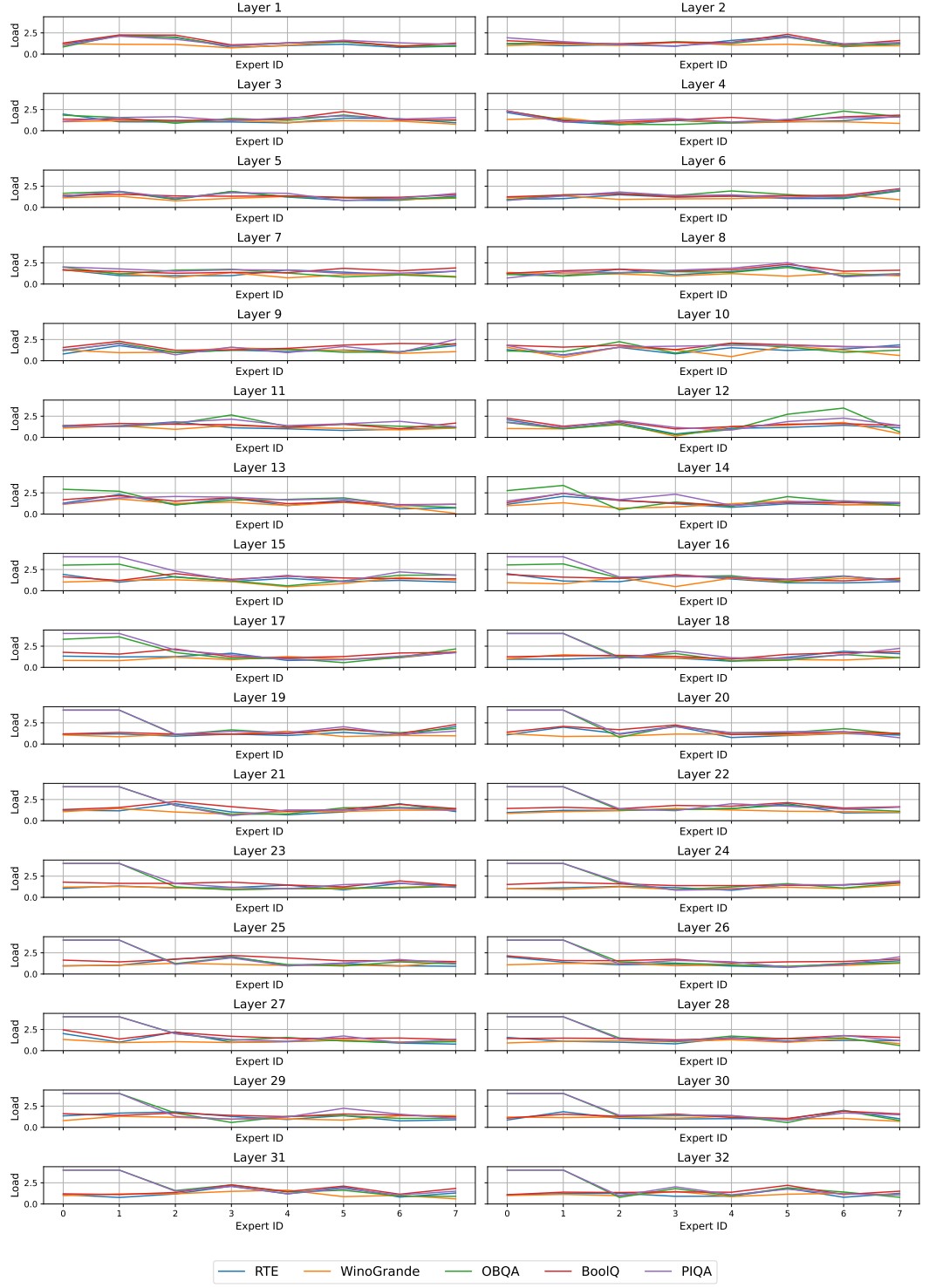

Figure 16: Layer-wise expert load in Mixtral-8×7B-Instruct.

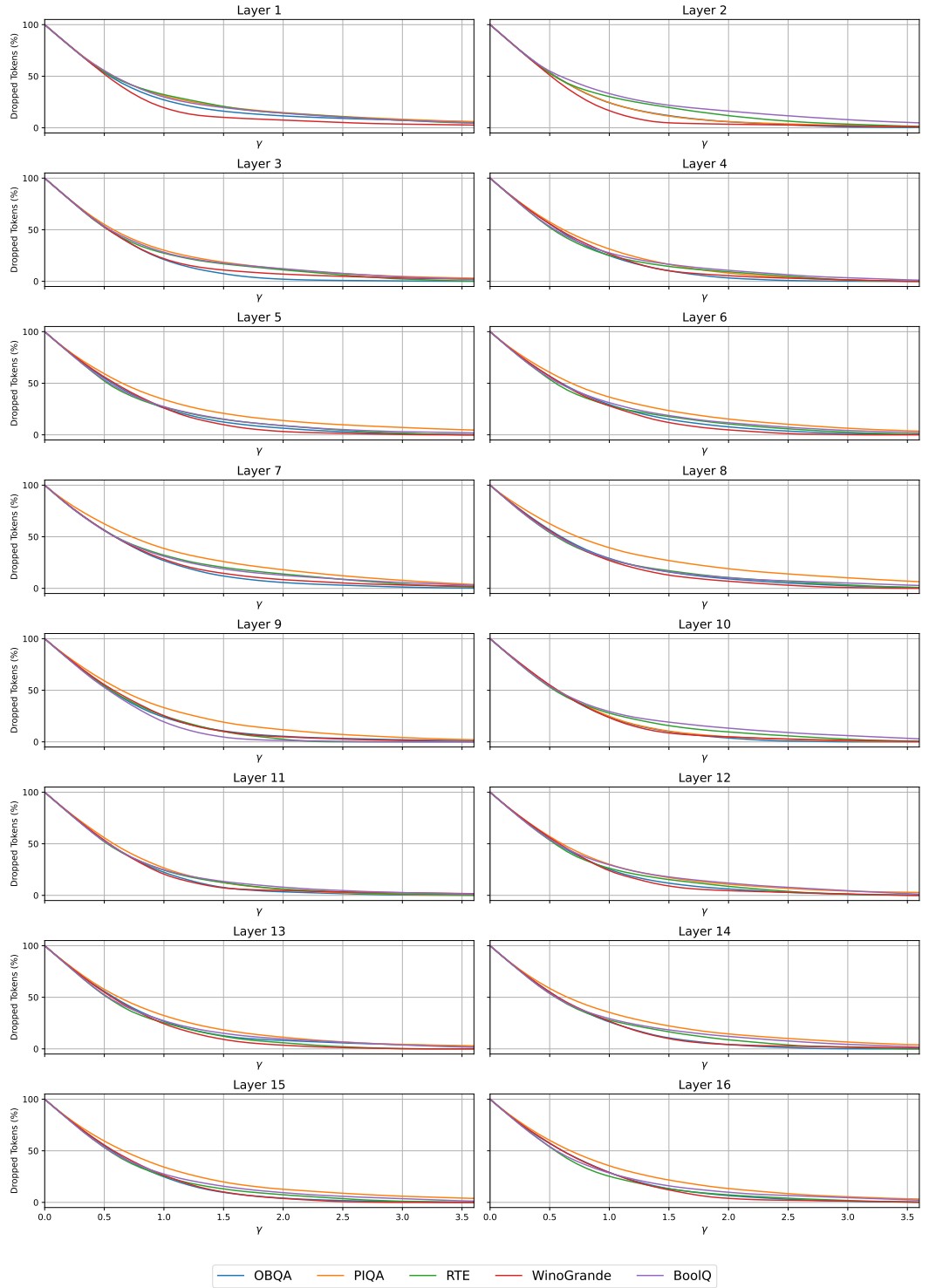

Figure 17: Dropped tokens with respect to capacity factors in OLMoE-Instruct.

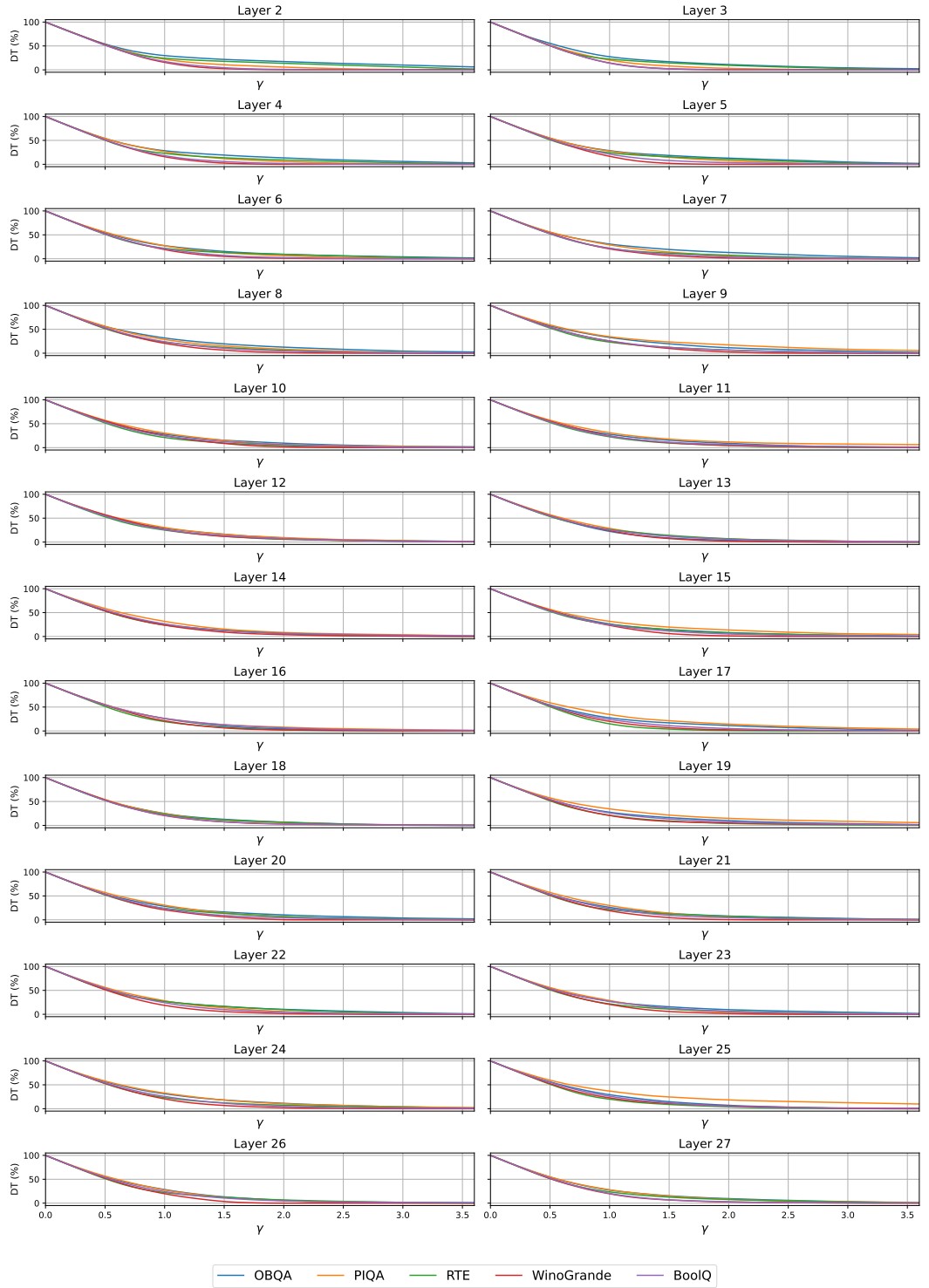

Figure 18: Dropped tokens with respect to capacity factors in DeepSeek-V2-Chat.

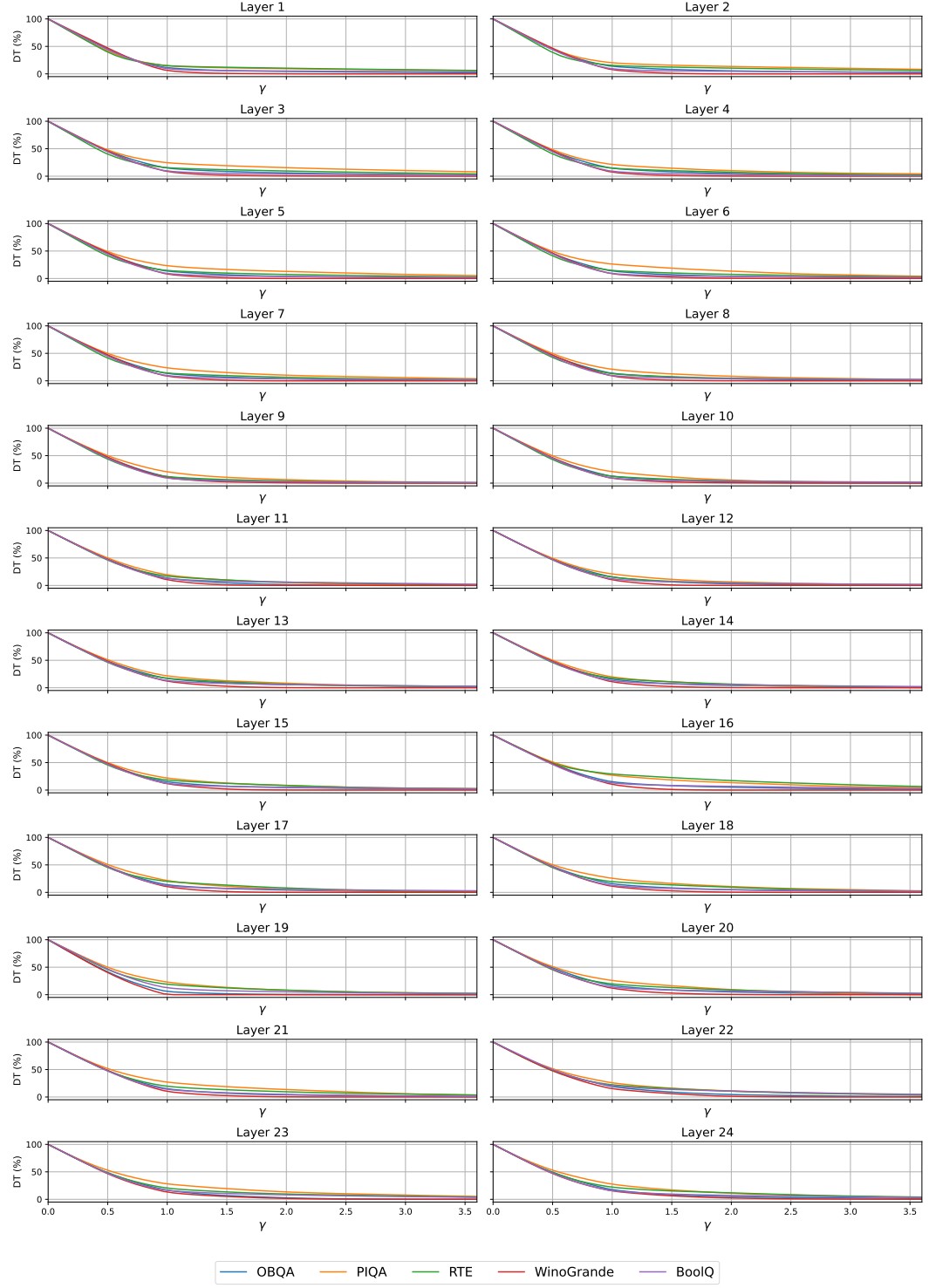

Figure 19: Dropped tokens with respect to capacity factors in Qwen1.5-MoE-Chat.

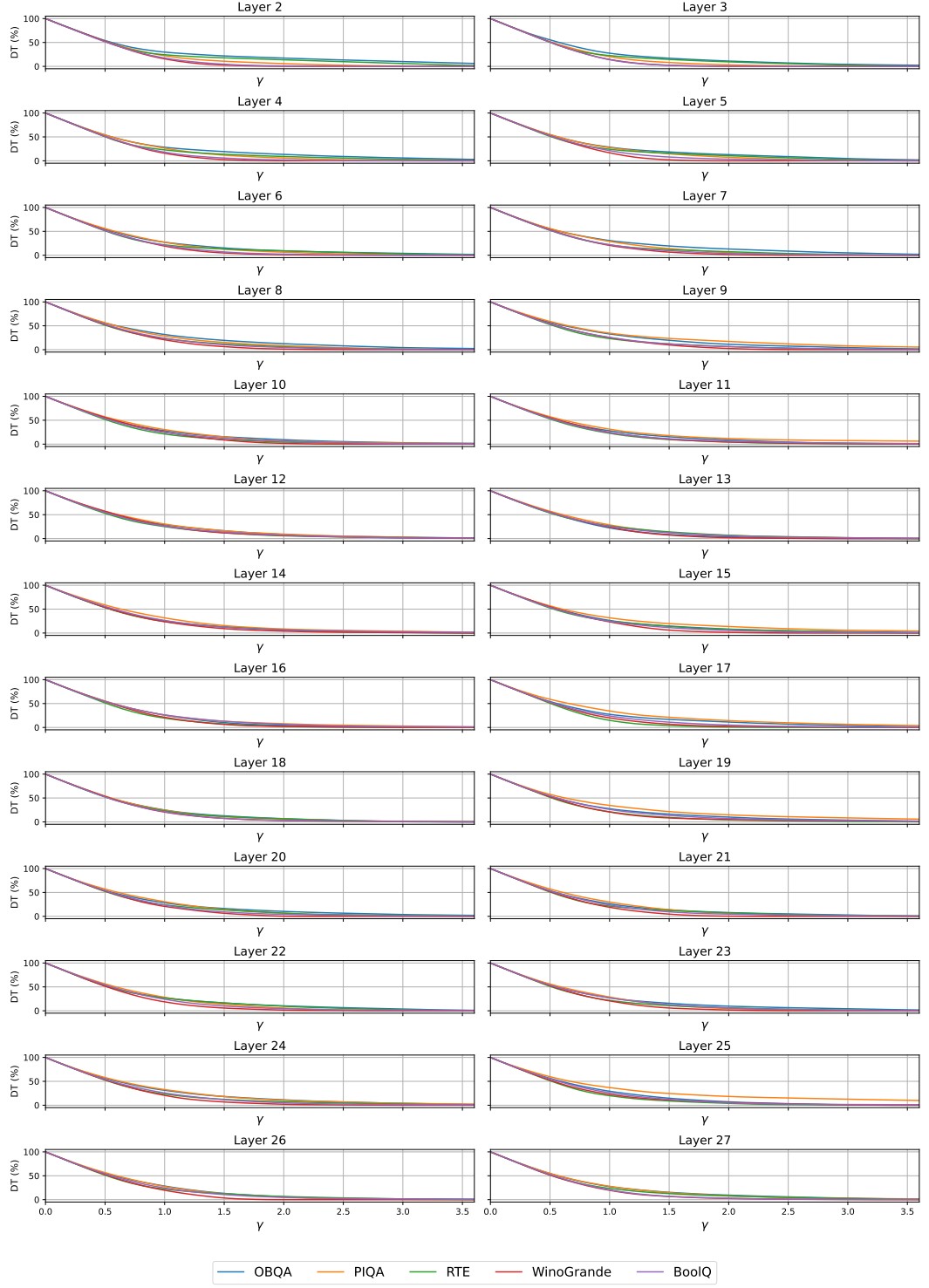

Figure 20: Dropped tokens with respect to capacity factors in Mixtral-8×7B-Instruct.

