# OpenReview forum: "Capacity-Aware Inference: Mitigating the Straggler Effect in Mixture of Experts"
_ICLR.cc/2026/Conference — ICLR 2026 Poster_

### Official Review · Reviewer_yGAq · 2025-10-30

**Soundness:** 3
**Presentation:** 3
**Contribution:** 2
**Rating:** 4
**Confidence:** 3

**Summary:**

The core problem addressed in this paper is the Straggler Effect in the inference phase of MoE systems: due to the imbalanced allocation between tokens and experts, overloaded experts become the bottleneck for inference latency, causing underutilized experts to remain idle while waiting, thereby reducing overall efficiency.
The key solution proposed is a Capacity-Aware Inference framework, consisting of Capacity-Aware Token Dropping and Capacity-Aware Expanded Dropping. The core concept involves discarding low-importance tokens for overloaded experts while increasing token allocation for underutilized experts. It achieves an average 1.85x inference speedup when applied to Mixtral-8×7B-Instruct.

**Strengths:**

1. The significance of the addressed problem lies in being the first to explicitly identify the "Straggler Effect" during MoE inference, and through quantitative analysis (e.g., revealing how some experts bear 7× the expected load N), it demonstrates its critical impact on inference latency.
2. The methodology is designed to be efficient: both strategies involve lightweight modifications during the inference phase, requiring neither adjustments to the model training process nor additional computational resources (such as the expert redundancy deployment in DeepSeek-V3). Token Dropping rapidly alleviates overload by discarding low-importance tokens, while Expanded Drop utilizes local expert expansion of candidate sets to avoid cross-device communication overhead.

**Weaknesses:**

1. The long-term impact of token dropping remains unvalidated: existing experiments only evaluate performance changes on static tasks (e.g., MMLU, OBQA), without investigating its effects on long-context generation (such as dialogue and text creation). For instance, whether persistent dropping of low-importance tokens may lead to degraded contextual coherence and generation quality remains unexamined, lacking verification in dynamic scenarios.
2. The method's generalizability has limitations: experiments show the approach achieves optimal results with models deploying "single expert per GPU" (e.g., Mixtral-8×7B-Instruct), but exhibits diminished acceleration effects for models with "multiple experts per GPU" (e.g., OLMoE-Instruct with 8 experts/GPU). The aggregated workload across multiple experts reduces the benefits of constraining straggler experts. The authors suggest "increasing GPU count to reduce experts per GPU" without proposing adapted solutions for multi-expert single-GPU scenarios, thus limiting universal applicability.
3. The study lacks comparison with existing inference optimization methods: it fails to benchmark the proposed approach against other MoE inference optimization solutions (such as PowerInfer's expert offloading or MoE-Infinity's activation-aware scheduling).

**Questions:**

1. Regarding the multiple experts per GPU scenario, the authors mention that aggregated workload diminishes the method's effectiveness but provide no specific improvement strategies. Have you considered dynamically adjusting the capacity limits for experts within a single GPU?
2. In the token dropping strategy, the gating score G(x) is used as the importance metric, but its reliability during inference has not been sufficiently validated. What about other important metrics (such as token's contextual contribution or semantic relevance), and how robust is the gating score across different tasks (e.g., logical reasoning, sentiment analysis)?
3. The experimental results show model inference speed improvements rely on reduced expert computation time, but fail to analyze changes in communication overhead. Does Expanded Drop's expansion of candidate expert sets increase token scheduling overhead within local devices?
4. Are there no related works on MoE inference acceleration? For example, https://github.com/LINs-lab/DynMoE

---

> ### Author Response · Authors · 2025-11-21
> **Response to Reviewer yGAq - Part I**
>
> We thank the reviewer for the thorough and well-articulated review. Your comments raise important questions and provide meaningful guidance for strengthening the work. We have carefully examined each point and provide our responses below.
>
> ---
>
> ### **W1. Long-term impact of token dropping remains unvalidated, e.g., dialogue and text creation**
>
> Thanks for your suggestions. We have conducted experiments on generation tasks (e.g., GSM8K in Table 2) to evaluate long-term effects, and we find that both Token Drop and Expanded Drop remain competitive. We also tested additional benchmarks such as HumanEval, NQ-Open, and MTS-Dialog, and observed consistent findings.
>
> #### **OLMoE-Instruct**
> | Model        | HumanEval | NQ-open | MTS-Dialog |
> |--------------|-----------|---------|------------|
> | Baseline     | 0.311     | 0.175   | 0.827      |
> | Token Drop   | 0.299     | 0.159   | 0.823      |
> | Expanded Drop| 0.303     | 0.170   | 0.829      |
>
> #### **Qwen1.5-MoE-A2.7B-Chat**
> | Model        | HumanEval | NQ-open | MTS-Dialog |
> |--------------|-----------|---------|------------|
> | Baseline     | 0.280     | 0.226   | 0.874      |
> | Token Drop   | 0.281     | 0.214   | 0.869      |
> | Expanded Drop| 0.291     | 0.224   | 0.872      |
>
> #### **DeepSeek-V2-Lite-Chat**
> | Model        | HumanEval | NQ-open | MTS-Dialog |
> |--------------|-----------|---------|------------|
> | Baseline     | 0.506     | 0.265   | 0.830      |
> | Token Drop   | 0.494     | 0.252   | 0.831      |
> | Expanded Drop| 0.508     | 0.268   | 0.831      |
>
> The capacity factor is set to 1.0. We report pass@1 for HumanEval, exact_match for NQ-Open, and BERTScore for MTS-Dialog.
>
> Motivated by your comments, we further evaluated DeepSeek-V2-Lite with capacity factor 1.0 on LongICLBench (with various prompt lengths). The results similarly show robust performance under long-context settings.
>
> | Prompt Length  | 2K  | 4K  | 7K  | 9K  | 11K |
> |------------------|-----|-----|-----|-----|-----|
> | **Baseline**      | 0.28|0.44|0.56|0.62|0.70|
> | **Token Drop**    | 0.27|0.43|0.57|0.65|0.71|
> | **Expanded Drop** | 0.23|0.53|0.63|0.62|0.70|
>
> We hope these results address your concern, and we will include them in the next version. As MoE layers process each token independently, long-range dependencies do not arise inside MoE layers themselves. Long contexts also contain substantial redundancy, so dropping a subset of tokens has limited impact on overall performance.
>
> ---
>
> ### **W2. The speedup of multi-expert on a single GPU**
>
> We appreciate the reviewer’s question. When multiple experts share a GPU, their workloads aggregate, reducing the proportion of reduced tokens and thus lowering overall speedup. This effect depends on the number of experts per device. As more GPUs are used and fewer experts reside on each GPU, this aggregation effect diminishes.
>
> To analyze this setting, we study a **device-level capacity constraint**, where:
>
> \[
> N_1 + N_2 + \cdots + N_m \le m \gamma \bar{N}
> \]
>
> Unlike expert-level constraints, device-level constraints only enforce the bound at the device level. Tokens may be dropped unnecessarily under expert-level constraints simply because one expert exceeds its limit. Device-level constraints admit more valid token assignments and thus are more flexible.
>
> #### **Evaluation Results (device-level vs expert-level)**
>
> | Setting        | GSM8K | HellaSwag | BoolQ | MMLU | OpenBookQA | PIQA | RTE | WinoGrande | AVG |
> |----------------|-------|-----------|-------|------|-------------|------|------|------------|------|
> | Baseline       | 89.5  | 77.6      | 88.6  | 77.9 | 45.2        | 80.6 | 82.3 | 69.5       | 76.4 |
> | Expert-γ=1.5   | 87.0  | 74.4      | 86.5  | 76.8 | 43.6        | 79.4 | 80.3 | 68.4       | 74.6 |
> | Device-γ=1.0   | 88.9  | 77.1      | 88.0  | 77.1 | 42.4        | 79.7 | 81.2 | 69.3       | 75.5 |
>
> Device-level constraints also yield stronger speedups. For example, on Qwen3-MoE with γ=1:
>
> - **1.31× end-to-end speedup**
> - **1.51× MoE-layer speedup**
>
> Both exceed the speedups under γ=1.5 (1.23× and 1.40× respectively).
>
> We will include this variant and its benefits in the updated draft.
>
> ---
>
> ### **W3. Comparison with existing inference optimization methods (e.g., PowerInfer)**
>
> We appreciate the reviewer’s suggestion. Our work focuses on test-time load balancing, while PowerInfer and similar methods optimize different aspects of inference—such as memory and I/O bottlenecks using GPU–CPU hybrid execution. Their optimization targets are therefore different from ours. Since the two approaches address orthogonal bottlenecks, they are **complementary**. Our method can be combined with systems like PowerInfer to further improve inference efficiency.

---

> ### Author Response · Authors · 2025-11-21
> **Response to Reviewer yGAq - Part II**
>
> ---
>
> ### **Q1. Dynamic adjustment of capacity limits within a single GPU**
>
> This is a great question. Dynamically adjusting capacities would require additional training, since it is unclear how to determine appropriate limits without learning them. Because our method is fully training-free, adding dynamic adjustments is non-trivial.
>
> However, we incorporate the idea through **device-level capacity**, which enforces token limits at the device level rather than per expert. Expert-level capacity is stricter and always processes fewer tokens, making device-level capacity a more flexible and practical variant. We now clarify this distinction clearly in the draft.
>
> ---
>
> ### **Q2. Alternative metrics vs. robustness of gating scores**
>
> Thanks for pointing this out. Contextual contribution or semantic relevance reflects global token importance, but our method drops **token–expert assignments**, not entire tokens. A token may be unimportant for some experts yet important for others. Thus, global metrics cannot capture fine-grained routing importance.
>
> Gating scores serve as an efficient, zero-overhead proxy, since they are already computed during routing. Their robustness comes from:
>
> - Low-score assignments contribute little; dropping them minimally affects outputs.
> - Each token selects k experts, so dropping some assignments still keeps multiple active experts.
> - The probability of a token losing all experts is extremely low (<1%).
> - Long-range semantics remain intact across tasks.
>
> We agree advanced metrics may further improve performance and consider this promising future work.
>
> ---
>
> ### **Q3. Scheduling overhead of Expanded Drop**
>
> Figure 6 shows the communication overhead (green bars). Expanded Drop is nearly identical to the baseline, since the only additional step is concatenating local top-k scores with scores from local experts. This cost is negligible in MoE inference.
>
> ---
>
> ### **Q4. Related works on MoE inference acceleration**
>
> We focus specifically on **test-time load balancing**, a topic with limited prior work. Existing MoE inference acceleration methods optimize different aspects (e.g., caching, routing, resource scheduling). In the revision, we will expand discussion of these works and clarify that they are orthogonal to our contributions.
>
> ---

---

> ### Author Response · Authors · 2025-11-27
> **Kind Reminder: Follow-up on Our Rebuttal Responses**
>
> Dear Reviewer yGAq,
>
> We sincerely appreciate the thoughtful feedback you provided earlier. Based on your suggestions, we have added new experiments, strengthened our analysis of the multi-expert per GPU setting, expanded the discussion on importance metrics, and clarified the relationship between our approach and existing MoE inference optimization work. These revisions have noticeably improved the clarity and completeness of the paper.
>
> If you have a moment to take another look at our responses, we would be grateful. Please let us know if there are any remaining points that would benefit from additional clarification. We are happy to refine the work further as the discussion moves toward its conclusion.
>
> Thank you again for the time and insight you have contributed to improving the paper.
>
> Best,
> Submission13559 Authors

---

### Official Review · Reviewer_vtDH · 2025-10-31

**Soundness:** 3
**Presentation:** 3
**Contribution:** 2
**Rating:** 4
**Confidence:** 5

**Summary:**

This paper identifies the **Straggler Effect** in Mixture-of-Experts (MoE) models during inference: overloaded experts cause global latency bottlenecks due to imbalanced token-to-expert assignments. To mitigate this, the authors propose **Capacity-Aware Inference**, introducing two lightweight strategies:

1. **Token Drop**: enforces capacity limits on overloaded experts by discarding low-importance tokens, reducing latency with minimal performance loss.
2. **Expanded Drop**: allows tokens to select additional local experts, improving utilization of underloaded experts and enhancing performance.

Experiments on both language and multimodal MoE models show significant improvements in inference efficiency (up to **1.85× speedup**) with comparable or even slightly better accuracy. The methods are training-free, easy to implement, and broadly applicable.

**Strengths:**

1. This paper is well written and organized, with clearly stated motivation and method, together with relatively comprehensive experiments.
2. This paper investigated both LLM and MLLM and demonstrated efficiency gains, demonstrating its effectiveness and universality.

**Weaknesses:**

1. The novelty of this paper is insufficient, since there has been comprehensive research on load balancing in MoE model training, such as Switch Transformer (Google) and AuxFree load balancing (DeepSeek). This paper does not have fundamental distinctions from these well-known papers, where the only difference is taming the load balancing issue during the inference stage in a training-free manner.
2. In Lines 322-323, you should clarify that DP is applied to attention modules while EP is applied to MoE modules.
3. You should specify the prefilling stage or decoding stage in your efficiency-related experiments with the listed workload, such as batch size and context length, which include Figures 4, 5, and 6.
4. Also, you should provide more comprehensive experimental results on efficiency benefits under different workloads. For example, you should scale the context length with a fixed batch size, and scale the batch size with fixed context length, to compare and visualize the speedup or latency, and therefore provide more insights with a scaled workload.

**Questions:**

1. What is the specific model you used in Figures 1 and 2? You should provide the detailed information in the caption texts.

---

> ### Author Response · Authors · 2025-11-21
> **Response to Reviewer vtDH**
>
> We thank the reviewer for the thorough and well-articulated review. Your comments raise important questions and provide meaningful guidance for strengthening the work. We have carefully examined each point and provide our responses below.
>
> ---
>
> ### **W1. Novelty of load balancing during test-time**
>
> While training-time load balancing has been extensively studied, test-time load balancing remains largely under-explored or is addressed only through approaches that require substantial additional resources. This gap is increasingly important as MoE models continue to gain popularity, while the computational resources available for serving them remain limited in most institutions.
>
> Moreover, training-time techniques do not guarantee balanced behavior at test time. We conduct comprehensive experiments across a diverse set of MoE models—covering different training pipelines (e.g., training from scratch vs. upcycling), different load-balancing strategies (with vs. without auxiliary balancing losses), and different architectural regimes (few experts vs. many experts). Despite these varied settings, all models exhibit significant test-time load imbalance, revealing a fundamental limitation of existing approaches and motivating the need for test-time solutions.
>
> In fact, more recent MoE architectures such as Qwen3-MoE not only fail to mitigate this issue but exhibit even more severe test-time imbalance. For example, the proportion of overflowed tokens reaches 46.1% for \(\gamma = 2.0\) and 57.0% for \(\gamma = 1.5\), compared to only 16.2% and 20.1% in OLMoE under the same settings.
>
> In summary, we respectfully disagree that prior training-time load-balancing approaches intersect with our contribution or undermine its novelty. These methods do not address the persistent and critical issue of test-time load imbalance, which remains prevalent in mainstream MoE models and is likely to persist. Our work systematically investigates this under-explored problem and introduces efficient, practical solutions explicitly designed for test-time deployment.
>
> ---
>
> ### **W2. Clarify that DP is applied to attention modules while EP is applied to MoE modules**
>
> Thanks for the suggestion. We have updated the draft to explicitly clarify that DP is applied to the attention modules, while EP is applied to the MoE modules.
>
> ---
>
> ### **W3. Clarification on the settings of efficiency-related experiments**
>
> Thanks for the suggestion. We have added the corresponding information to Figures 4, 5, and 6, and we now explicitly include the batch size and sequence length in the descriptions.
>
> The straggler effect becomes more significant when GPUs are highly utilized; otherwise, GPUs retain spare compute capacity and maintain high throughput, making speedup differences less meaningful. For this reason, the results reported in the paper are measured under high-utilization conditions. In our experiments, this corresponds to the batch size of 8k and the sequence length of 512.
>
> which reflects a realistic server-side deployment where a large number of queries arrive concurrently.
>
> ---
>
> ### **W4. More comprehensive experimental results under different workloads**
>
> Following your suggestion, we conducted experiments under varying batch sizes and prompt lengths. Below are additional results on DeepSeek-V2 with \(\gamma = 2.0\):
>
> | Batch Size | Prompt Length | Speedup |
> |-----------:|--------------:|--------:|
> | 8k         | 100           | 1.09    |
> | 8k         | 200           | 1.18    |
> | 8k         | 400           | 1.24    |
> | 4k         | 1k            | 1.26    |
> | 2k         | 1k            | 1.27    |
> | 2k         | 2k            | 1.27    |
> | 1k         | 1k            | 1.27    |
> | 1k         | 2k            | 1.24    |
> | 1k         | 4k            | 1.23    |
>
> These results indicate that under heavier workloads, GPUs operate at higher utilization with less spare capacity, making the speedup more pronounced. Under lighter workloads, GPUs can absorb most of the computation, resulting in smaller speedup. In realistic server-side MoE deployment, the workloads are substantially higher, and our techniques effectively mitigate the resulting straggler effect.
>
> We will incorporate these new results into the revised version.
>
> ---
>
> ### **Q1. The specific model used in Figures 1 and 2**
>
> Both Figure 1 and Figure 2 use OLMoE.
> We have added the note *“Example shown with OLMoE on OpenBookQA”* to the caption of Figure 1.
> For Figure 2, the model (OLMoE) is already specified in line 159, and we have updated the caption to make this clearer.
>
> ---

---

> > ### Author Response · Authors · 2025-11-27
> > **Follow-up Reminder to Reviewer vtDH**
> >
> > Dear Reviewer vtDH,
> >
> > We sincerely appreciate the detailed and constructive feedback you provided earlier. Based on your suggestions, we have added new experiments under varied workloads, clarified the efficiency-experiment settings, expanded the discussion of test-time load balancing versus training-time methods, and improved the descriptions in Figures 1 and 2. These revisions have noticeably strengthened both the clarity and the completeness of the paper.
> >
> > If you have a moment to revisit our responses, we would be grateful. Please feel free to let us know if any points would benefit from additional clarification or if there are aspects you believe could be refined further. We are happy to continue improving the paper as the discussion progresses.
> >
> > Thank you again for your careful review and for the insights that helped us substantially improve the work.
> >
> > Best,
> > Submission13559 Authors

---

### Official Review · Reviewer_TfSK · 2025-11-01

**Soundness:** 2
**Presentation:** 2
**Contribution:** 2
**Rating:** 6
**Confidence:** 3

**Summary:**

This paper identifies that Mixture-of-Experts (MoE) models exhibit highly imbalanced token-to-expert assignments during inference, leading to inefficient inference computation. To address this, the authors propose Capacity-Aware Token Drop, where overloaded experts drop excess tokens beyond the compputation budget. To further compensate for potential performance loss and utilize idle capacity on underloaded experts, they introduce Capacity-Aware Expanded Drop, allowing tokens to leverage additional underutilized experts on the same device. Experiments across multiple models demonstrate that the proposed methods achieve notable inference speedups with minimal or no performance degradation.

**Strengths:**

* The experimental section is rich and goes beyond reporting results—it includes several insightful analyses that provide independent value.

* The proposed approach is simple  and easy to implement.

**Weaknesses:**

* When the number of experts per device exceeds one, the observed speedup becomes limited.

* The writing could be clearer in several places (see “Questions” below).

**Questions:**

## Writing clarify

* Figure 1: The definition of "normalized load" should be explained so that the figure is self-contained; it’s not clear until Section 3.

* Which dataset and model are used in Figure 1?

* Line 204 mentions that the capacity constraint is applied per device rather than per expert. However, later discussion (e.g., Line 217: "each overflowed expert selectively discards those with lower scores") seems to describe per-expert constraints. This discrepancy makes the algorithm unclear.

* Token Drop: If I understand correctly, a token could potentially end up with no assigned expert after dropping, if its scores assigned to each expert are ranked below the threshold. How does the algorithm ensure each token still has a valid output?

* Expanded Drop: It seems that each token additionally selects local experts (beyond the normal top-k experts), and these experts then drop tokens following the same rule as in "Token Drop Regulates the Latency of High-Load Experts" section. In that case, could an expert retain some tokens assigned as local experts while dropping others from the top-k assignment? Please clarify.

* Does Table 1 include the Expanded Drop technique?

* In Figure 5, how many experts per GPU are used for models other than Mixtral?

* In Figure 6, do the x-axis labels "1.0," "1.5," and "2.0" correspond to the \gamma values?

* In Figure 9, which model and dataset are used?

* Line 448 states that "we also experiment with a strategy that prioritizes dropping image tokens before selectively removing text tokens (“Image First”)." Could the authors explain the implementation details of this strategy?

## Experiments

* Eq. 4 resents the latency of an MoE layer. The intuition is sound, but latency in practice depends on many factors. It would strengthen the paper to:
    * (1) Measure actual layer latency and compare it with this theoretical bound; and
    * (2) Measure end-to-end model latency to verify whether max({N_i}i=1^n) remains the dominant factor.

* Line 340: The claim that "As illustrated in Figure 5, for Mixtral-8×7BInstruct, deploying a single expert per GPU maximizes the effectiveness of capacity-aware inference." requires experiments with multiple experts per GPU for the Mixtral model. Currently, Figure 5 only reports results for a single expert per GPU for the Mixtral model.

* Line 352 states that "Notably, the duration of permutation and communication increases when tokens are expanded across a range of global experts." Could the authors clarify what it means and how to see it from Figure 6.

* Line 373: The paper claims "regulating the maximum capacity has a limited impact on the overall number of accommodated tokens," yet Figure 7 shows that varying \gamma dramatically changes the percentage of dropped tokens from nearly 100% to 0%. Please clarify.

* Figure 3: Why does “Expanded Drop + Image First” significantly outperform the baseline? Additional explanation would be helpful.

## Other minior issues

* The citation format is incorrect in many places. For example, \citep should be used in:
    * "In recent years, the rapid evolution of Large Language Models (LLMs) OpenAI (2024); Team (2024a); DeepSeek-AI et al. (2024b) ..."
    * "Specifically, MoE Shazeer et al. (2017b); Fedus et al. (2022) enhances scalability ..."
    * "We conduct experiments on OLMoE Muennighoff et al. (2024), Qwen1.5-MoE Team (2024b), DeepSeek-V2-Lite DeepSeek-AI et al. (2024a) and Mixtral Jiang et al. (2024)"
    * "The number of shots for each task is detailed in Table 4, which includes multiple tasks: ARC-C Clark et al. (2018), BoolQ Clark et al. (2019), HellaSwag Zellers et al. (2019), MMLU Hendrycks et al. (2021), OBQA Mihaylov et al. (2018), PIQA Bisk et al. (2019), RTE Wang et al. (2019), WinoGrande ai2 (2019) and GSM8K Cobbe et al. (2021). The evaluation code is based on EleutherAI’s LM Harness framework Gao et al. (2023)."

* Section 1 claims the identification of "the Straggler Effect caused by token imbalance at inference time in Mixture of Experts" as a contribution. However, as discussed in the paper, this phenomenon is already mentioned in prior work (e.g., DeepSeek-V3). It should be framed as a motivation rather than a contribution.

* Line 706: Add a comma before "and"

* Line 713: Add a comma before "and"

---

> ### Author Response · Authors · 2025-11-21
> **Response to Reviewer TfSK - Part I**
>
> Thanks for your recognition of our work and for the detailed suggestions on how to further improve it. Our updates are as follows:
>
> ---
>
> ### **W1. The observed speedup when the number of experts per device exceeds one**
>
> We acknowledge that when aggregating loads from multiple experts (including both high-load and low-load experts), the proportion of reduced load becomes smaller, which in turn leads to lower speedup. This is because the experts on the same device do not run fully in parallel. However, this effect diminishes as the number of GPUs increases and fewer experts reside on each individual GPU.
>
> Another potential technique to improve efficiency in this setting is to apply a **device-level capacity constraint**, which limits the total capacity across all experts on the same device. In this case, we constrain the sum of assigned tokens on each device, for example
> \[
> N_{1} + N_{2} + \dots + N_{m} = m \gamma \bar{N}.
> \]
> This provides a smoother and less restrictive constraint than expert-level capacity. Under expert-level capacity, tokens may be dropped even when
> \[
> N_{1} + N_{2} + \dots + N_{m} < m \gamma \bar{N},
> \]
> simply because some individual experts exceed their limits. In contrast, device-level capacity admits these tokens as long as the total device-level capacity is not violated.
>
> Consequently, under the same capacity factor, device-level Expanded Drop achieves higher performance than expert-level Expanded Drop, as shown below:
>
> | Setting          | GSM8K | HellaSwag | BoolQ | MMLU | OpenBookQA | PIQA | RTE  | WinoGrande | AVG  |
> |------------------|-------|-----------|-------|------|-------------|------|------|------------|------|
> | Baseline         | 89.5  | 77.6      | 88.6  | 77.9 | 45.2        | 80.6 | 82.3 | 69.5       | 76.4 |
> | Expert-γ = 1.5   | 87.0  | 74.4      | 86.5  | 76.8 | 43.6        | 79.4 | 80.3 | 68.4       | 74.6 |
> | Device-γ = 1.0   | 88.9  | 77.1      | 88.0  | 77.1 | 42.4        | 79.7 | 81.2 | 69.3       | 75.5 |
>
> This also shows that device-level capacity can operate effectively under lower capacity factors, further enhancing the efficiency of capacity-aware inference. For example, on Qwen3MoE with device-level \(\gamma = 1.0\), the end-to-end speedup is **1.31×** and the single MoE layer speedup is **1.51×**, both higher than the **1.23×** end-to-end speedup and **1.40×** single-layer speedup under expert-level \(\gamma = 1.5\).
>
> We will treat device-level capacity-aware inference as a variant of expert-level capacity inference and clarify its contributions in scenarios with multiple experts on a single device. If you have any additional suggestions, please let us know.
>
> ---
>
> ### **W2. The writing could be clearer in several places.**
>
> Thanks for your comments. We have carefully considered your suggestions and made the corresponding revisions in the draft.
>
> ### **Q1. Normalized load in Figure 1**
>
> Thanks for your suggestion. We have added a description of normalized load in the caption of Figure 1. Specifically, the normalized load is computed as each expert’s load divided by the mean load across all experts.
>
> ---
>
> ### **Q2. Dataset and model used in Figure 1**
>
> OpenBookQA and OLMoE are used in Figure 1.
> We have included the note “Example shown with OLMoE on OpenBookQA” in the caption.
>
> ---
>
> ### **Q3. Clarification on Line 204 about the implementation of capacity constraint**
>
> Thank you for pointing this out. Token Drop is indeed applied within each expert. Line 204 is meant to clarify that Token Drop operates on the tokens currently residing on each device: excessive tokens are dropped before the remaining tokens are dispatched to their target experts. We have updated the draft accordingly.
>
> ---
> **References**:
> [1] *Mixture-of-Depths: Dynamically allocating compute in transformer-based language models*, Raposo et al.
> [2] *Towards Efficient Mixture of Experts: A Holistic Study of Compression Techniques*, He et al.

---

> ### Author Response · Authors · 2025-11-21
> **Response to Reviewer TfSK - Part II**
>
> ### **Q4. The extreme situation where the scores assigned to each expert are ranked below the threshold**
>
> This is a good question. In principle, it is possible that a token ends up with no assigned experts after Token Drop, but the probability is extremely low. As shown in Figure 7, the proportion of dropped token–expert assignments is relatively small (e.g., around 30% under a capacity factor of 1.0). On average, each token–expert mapping has approximately a 30% probability of being dropped. Since each token selects \(k\) experts, the probability that none of them remain after dropping is roughly \((0.3)^k\) (assuming a simplified independent and identically distributed pattern; the real distribution differs slightly). This value becomes negligible as \(k\) increases.
>
> In practice, we also measure this probability using OLMoE-Instruct on Bank77 over 5 rounds in LongICLBench. The observed ratio is about **0.67%**. Although real-world data does not strictly follow the i.i.d. assumption, this confirms that such cases are exceedingly rare.
>
> Moreover, not all tokens are equally important. Skipping less informative tokens can preserve model performance, as shown in prior work [1]. Similarly, dropping tokens in certain MoE layers does not substantially degrade accuracy. For example, [2] removes all tokens in selected layers of Mixtral while still retaining around 90% of the original performance on some tasks. In contrast, our methods are more stable because we neither remove entire layers nor drop all tokens in any layer, which leads to significantly higher performance.
>
> ---
>
> ### **Q5. Possible situation where an expert retains some tokens assigned as local experts while dropping others from the top-k assignment**
>
> This situation can occur, but only with very low probability. It happens when the top-\((k+m)\) expert scores of some tokens exceed the top-\(k\) scores of other tokens. In such cases, the latter tokens effectively rely on only a few dominant experts (fewer than \(k\)), while the former tokens exhibit flatter expert distributions. Consequently, this behavior strengthens the representations of tokens with flatter expert preferences while removing low-importance token–expert mappings that contribute little to the overall output.
>
> In practice, this replacement happens extremely rarely. For example, only **0.16%** of such cases occur for OLMoE-Instruct on Bank77 in LongICLBench.
>
> ---
>
> ### **Q6. Explanation of Table 1**
>
> Table 1 is only related to Token Drop, and it includes a discussion of dropping metrics. It is not associated with Expanded Drop.
>
> ---
>
> ### **Q7. Number of experts per GPU in models other than Mixtral (Figure 5)**
>
> We deploy 8 experts per GPU for OLMoE-Instruct and DeepSeek-V2-Lite, and 10 experts per GPU for Qwen1.5-MoE-Chat.
> This setup is practical given our available resources and ensures that the total number of experts is divisible by the number of experts per GPU.
> This information is already stated in Lines 343–345 of the current draft.
>
> ---
>
> ### **Q9. X-axis labels in Figure 6 (“1.0”, “1.5”, “2.0”)**
>
> Yes. Since no additional factors are involved, we use the abbreviated numeric values for simplicity. We will incorporate your suggestion and clarify this explicitly in the revised version.
>
> ---
>
> ### **Q10. The model and dataset used in Figure 9**
>
> Figure 9 evaluates **OLMoE** on the **OpenBookQA** dataset.
>
> ---
>
> ### **Q11. Implementation details of the “Image First” strategy**
>
> Thanks for the question. In the Image-First strategy, we first drop image tokens to satisfy the capacity constraint. In most cases, this step is sufficient, as shown in Figure 10, where the majority of tokens are image tokens. If the constraint is still not met after dropping image tokens, we then proceed to drop text tokens.
>
> ---
> **References:**
> [1] *Mixture-of-Depths: Dynamically allocating compute in transformer-based language models*, Raposo et al.
> [2] *Towards Efficient Mixture of Experts: A Holistic Study of Compression Techniques*, He et al.

---

> ### Author Response · Authors · 2025-11-21
> **Response to Reviewer TfSK - Part III**
>
> ### **Q12. Discussion of factors affecting latency and latency measurement**
>
> You are right that latency arises from multiple components. Equation 4 provides an intuitive characterization of MoE-layer latency, but it is not intended to serve as a strict deterministic formula. In Figure 6, we decompose the latency into gating, expert computation, perturbation, and communication. Our method primarily mitigates the straggler effect by preventing excessively high-load expert computation and by reducing communication cost through early token dropping.
>
> We have already conducted both measurements suggested by the reviewer:
> **(1)** *Actual MoE-layer latency:* Figure 4 reports measured layer-level latency under different capacity factors and clearly shows that reducing \(\max(N_i)\) lowers the real latency of MoE layers.
> **(2)** *End-to-end model latency:* Figure 5 presents the end-to-end latency across the full model and demonstrates that \(\max(N_i)\) remains an important factor even after accounting for attention and other non-MoE components.
>
> We also experiment with different capacity factors, which directly control \(\max(N_i)\), allowing us to observe how these changes affect both layer-level and end-to-end latency. These empirical findings confirm the practical relevance of Equation 4.
>
> ---
>
> ### **Q13. Reporting results for a single expert per GPU only for Mixtral**
>
> Thanks for this question. The experimental setting is primarily constrained by our lab’s available resources. For example, deploying OLMoE with a single expert per GPU would require 64 GPUs, which is beyond our capacity. However, the trend remains clear: configurations with one expert per GPU exhibit larger speedups because the load imbalance across devices becomes more pronounced. In contrast, when multiple experts share the same GPU, the workloads of high-load and low-load experts tend to aggregate, making the imbalance less significant and reducing the achievable speedup.
>
> ---
>
> ### **Q14. Discussion of expanding global experts vs. local experts**
>
> We evaluate Expanded Drop in two settings:
> • **Local expansion**, which extends the candidate expert set only to the \(m\) local experts on the same device.
> • **Global expansion**, which expands to the top-\((k+m)\) experts across all devices.
>
> The global variant requires cross-device communication, introducing perturbation and noticeably increasing communication overhead, as reflected in Figure 6. Therefore, we recommend the local version, which preserves efficiency without incurring additional communication costs.
>
> ---
>
> ### **Q15. Impact of capacity constraints on the number of accommodated tokens**
>
> Thanks for this question. We report the ratios of dropped tokens across different capacity settings, including the extreme case of \(\gamma = 0\) for completeness. When \(\gamma = 0\), it is expected that all tokens are dropped. However, in our actual experiments, we use capacity factors of 1.0, 1.5, and 2.0. Under these practical settings, the proportion of dropped tokens remains small, so the impact on the overall number of processed tokens is limited.
>
> ---
>
> ### **Q16. Why “Expanded Drop + Image First” significantly outperforms the baseline**
>
> Image tokens exhibit much higher redundancy than text tokens, as observed in prior work such as FastV [1]. Although image sequences contain many tokens, only a small subset is needed to preserve essential visual information. In contrast, text tokens carry more direct semantic meaning used for instructions and reasoning, so aggressively dropping text tokens (e.g., using a capacity factor of 0.5) can harm instruction-following ability.
>
> Leveraging this insight, we apply the Image-First strategy and find that multimodal tasks can use much lower capacity factors without degrading performance, since image tokens contain significantly more redundancy.
>
> ---
>
> ### **Q17. Citation format**
>
> Thanks for pointing this out. We have fixed the citation formatting in the new version of the draft.
>
> ---
>
> ### **Q18. Using the Straggler Effect as the motivation**
>
> Thanks for pointing this out. Our intention was to provide an initial qualitative analysis of the test-time straggler effect and make the phenomenon more explicit to the community. We also acknowledge that this effect serves as an important motivation for our work. We have refined the corresponding description in the updated draft.
>
> ---
>
> ### **Q19. Line 706 and Line 713: Add comma before “and”**
>
> Thanks for your advice. We have added the commas in the corresponding lines.
>
> ---
>
> ### **Reference**
>
> [1] *An Image is Worth 1/2 Tokens After Layer 2: Plug-and-Play Inference Acceleration for Large Vision-Language Models*, Chen et al., ECCV 2024 (Oral).

---

> > ### Author Response · Authors · 2025-11-25
> > **Kind Reminder: Follow-up on Our Rebuttal Responses**
> >
> > Dear Reviewer TfSK,
> >
> > Thank you again for your thoughtful and detailed feedback. We have carefully addressed every point you raised, including additional analyses, new experimental results, and clarifications throughout the draft. Our full responses are provided above.
> >
> > If you have a moment, we would greatly appreciate it if you could take a look at our updates and let us know whether the revisions address your concerns. Your follow-up feedback would be very helpful as we refine the final version of the work.
> >
> > Thank you again for your time and constructive input. It has truly helped strengthen the paper.
> >
> > Best regards,
> > Submission13559 Authors

---

> ### Comment · Reviewer_TfSK · 2025-11-28
>
> Thank you for the response. I have the following follow-up questions:
>
> **Q4**: *"we also measure this probability using OLMoE-Instruct on Bank77 over 5 rounds in LongICLBench. The observed ratio is about 0.67%."* When this situation occurs, how do we process tokens that fail to pass any expert? What output should be produced for them?
>
> **Q12**: Figures 4 and 5 show results *after* your method is applied, whereas Equation 4 appears in the motivation section, before the method is introduced. Therefore, my original suggestion was to illustrate how `max({N_i}_{i=1}^n)` affects the layer-wise or end-to-end latency in a *vanilla* MoE model, before applying your method.
>
> **Q13**: I am *not* asking you to run “a single expert per GPU” for more models. Let me clarify my questions again. To support your claim "As illustrated in Figure 5, for Mixtral-8×7BInstruct, deploying a single expert per GPU maximizes the effectiveness of capacity-aware inference.", we need to show Mixtral-8×7B-Instruct with multiple choices of the number of experts per GPU—for example: one expert per GPU, two experts per GPU, four experts per GPU, etc. However, Figure 5 only reports results for a single-expert-per-GPU configuration for the Mixtral model.
>
> **Q16**: Let me clarify my question again. I understand why the Image-First strategy is better *without degrading performance*, but the surprising phenomenon is that it actually *outperforms* the baseline by a significant margin. Why does this happen?
>
> **Overall**: I also noticed that many of the clarifications provided in the rebuttal are still not reflected in the revision. For example, the issues raised in Q15, Q9, and Q18 remain unaddressed (e.g., for Q18, “We identify the Straggler Effect caused by token imbalance at inference time in Mixture of Experts” is still listed as the first contribution (line 87 in the revision)). I would recommend incorporating all of these clarifications in the rebuttal into the paper to reduce similar confusion for future readers.

---

> ### Author Response · Authors · 2025-11-30
> **Further Response to Reviewer TfSK**
>
> We sincerely appreciate the reviewer’s thoughtful follow-up questions and the opportunity to clarify these points. We provide detailed responses below.
>
> ### **Q4. How are tokens processed and what is the output when no expert is selected.**
>
> In Equation 2, the MoE layer output for a token is computed as a weighted sum over the outputs of its selected experts. In the extremely rare case that a token ends up with no selected experts after Token/Expanded Drop, then by the MoE formulation, no expert output contributes to that token at this layer. As a result, the model naturally skips the MoE transformation for this token, and its hidden state simply passes through unchanged via the residual connection.
>
> Because only a tiny fraction of tokens bypass the MoE layer, the impact on overall performance is minimal. Notably, this behavior is also far less aggressive than prior work such as dropping all image tokens [1] or removing all tokens in selected MoE layers [2], both of which still maintain strong performance in certain settings.
>
> ---
>
> ### **Q12. Discussion of how max(\{N_i\}_{i=1}^n) affects layer-wise and end-to-end latency**
>
> We do illustrate how max(\{N_i\}_{i=1}^n) affects MoE latency in the paper. Lines 48–53 and 164–175 describe the inference pipeline of MoE models in distributed systems and how load imbalance leads to the straggler effect, where the most heavily loaded experts dictate the overall latency of the MoE layer. This is a phenomenon in vanilla MoE models and directly motivates our exploration of mitigating this issue.
>
> Equation 4 is intended to capture this intuition, and Figures 4 and 5 are consistent with it. These figures report the efficiency of MoE models before and after deploying our methods under different capacity factors. Since the capacity factor effectively regulates the maximal expert load (max(\{N_i\})), the results in Figure 4 and Figure 5 empirically demonstrate the relationship between maximal load and latency.
>
> Equation 4 also motivates the definition of the capacity factor and the design of our proposed methods. For this reason, we believe it is reasonable to present Equation 4 before the method and experimental sections. That said, we will continue to refine the exposition in the revised version to make this motivation clearer to readers.
>
> ---
>
> ### **Q13. Multi-expert-per-GPU results for Mixtral-8×7B**
>
> Thanks for the clarification. We agree that examining Mixtral-8×7B under multiple experts-per-GPU configurations strengthens our claim.
>
> Following your suggestion, we attempted to evaluate Mixtral under several multi-expert-per-GPU settings. However, placing four or eight experts per GPU exceeds the memory budget under our evaluation configuration (batch size, sequence length, and etc.), and the tests could not be completed due to CUDA out-of-memory errors.
>
> We were able to successfully run the two-experts-per-GPU configuration, which fits within memory constraints. Under this setting, Mixtral achieves 1.54× and 1.51× end-to-end speedups with Token Drop and Expanded Drop, respectively.
>
> We have added these results to Figure 5 in the revised draft to provide a more complete analysis across different experts-per-GPU configurations.
>
> ---
>
> ### **Q16. The competitive performance of the Image-First strategy**
>
> Expanded Drop with the Image-First strategy can outperform the baseline in some multimodal tasks, but this is not for all settings. As shown in Table 6 and Figure 11, under aggressive capacity factors such as 0.5, the performance in certain cases is comparable to or slightly below the baseline.
>
> The competitive performance can be explained from two perspectives:
>
> 1. **High redundancy in image tokens.**
>    Prior work such as FastV shows that a large portion of image tokens are redundant, and removing them can even improve robustness by reducing noisy or non-informative visual signals. Our findings are consistent with these observations. By prioritizing the dropping of redundant image tokens, Expanded Drop effectively reduces noise while retaining the key visual features required for the task.
>
> 2. **More effective expert utilization under Expanded Drop.**
> Expanded Drop maintains the representation capability: even after removing low-importance token–expert assignments, most tokens still preserve multiple meaningful expert paths. This avoids skipping entire layers, which poses a much higher risk of losing important transformations (as in methods that drop all tokens in certain layers). As a result, EP can in some cases produce stronger representations than the baseline routing.
>
> These two factors together explain why the Image-First variant of Expanded Drop can match, and in some cases even outperform, the baseline.
>
> ---
>
> ### **Incorporating these clarifications**
>
> Thanks again for these helpful suggestions. We have followed your recommendations in the revised version and will continue to incorporate these clarifications in the paper to make it clearer.

---

### Official Review · Reviewer_yZTe · 2025-11-11

**Soundness:** 4
**Presentation:** 3
**Contribution:** 3
**Rating:** 8
**Confidence:** 4

**Summary:**

The authors propose a method for reducing the inference time of mixture-of-experts (MoE) models by load balancing during inference. In MoE models, tokens are typically routed to top-k most relevant experts and uneven loads across experts significantly slows down inference since sync operations are performed after the slowest expert has completed (termed ‘straggler effect’). The work proposes dropping tokens associated with overloaded experts and potentially re-routing them to underloaded ones to reduce the peak load on any single expert. Experiments on several MoE models and benchmark datasets suggest that the proposed method results in huge speedups with minor drop in performance.

**Strengths:**

The paper is mostly well written and easy to understand. The authors identify a specific issue with MoE models and propose a solution to address it. The idea is simple, interesting and well motivated. The experiments are thorough, with results on multiple models, benchmark datasets and ablations and analysis to understand the different components of the proposed method. The authors show that existing models suffer from the ‘straggler effect’ and that the proposed solution can alleviate it, achieving 7-18% speedup with up to 0.4% points drop in average performance. The solution is also extended to multi-modal models, where huge gains can be achieved by dropping the redundant tokens in the image domain.

**Weaknesses:**

The work does not have any major weaknesses.

1. There is not much discussion/comparison with other approaches to address the ‘straggler effect’. For instance, the authors mention the use of additional GPUs for overloaded experts. For commercial use-cases where there are thousands/millions of concurrent requests to a service, would just having extra servers for overloaded experts be efficient or would the proposed solution still be necessary given that it comes with a performance drop? The proposed method is also shown to provide the biggest improvements when the number of experts per GPU is reduced by increasing the number of GPUs.
2. Discussion on the effect of different sequence lengths within a batch and dataset is missing. Since sequences in a batch can be of different lengths and the proposed method prunes tokens at the batch level, how does the variance in length affect efficiency and performance? Also, how would a high variance in sequence lengths within a dataset affect the speed and performance?
3. The expanded token drop method (in the Methodology section) is a bit unclear. How exactly are the additional ‘m’ experts chosen and how are they ensured to be ‘local’ experts?
4. Since some experts specialize towards particular domains, the proposed method might have a bigger drop in performance on specific tasks/datasets. For instance, in table 2, the maximum drop in performance over the 8 datasets on DeepSeek-V2-Lite-Chat model is 2.4% points while the average drop is 0.4% points (although this seems to be a one-off scenario in the table). How can this be predicted / handled?

**Questions:**

Please refer to the weakness section.

---

> ### Author Response · Authors · 2025-11-21
> **Response to Reviewer yZTe**
>
> We are grateful for your positive and encouraging feedback. Thank you for the valuable comments and suggestions. We have carefully reviewed each point and provide our detailed responses below.
>
> ---
>
> ### **W1: The discussion of approaches toward the ‘straggler effect’ and the broad application of proposed methods.**
>
> Thanks for your comment. Our paper already discusses DeepSeek’s use of additional experts to alleviate the straggler effect during inference. While adding experts can indeed mitigate this issue, it requires deploying extra servers. In practice, however, computational resources are often limited and shared across multiple tasks in the queue. In such cases, we advocate for more resource-efficient solutions that address the straggler effect without relying on additional servers.
>
> Capacity-aware inference maintains performance within an acceptable range and is orthogonal to simply adding more GPUs. Moreover, scaling to more GPUs reduces the number of experts per device, which exacerbates the device-level straggler effect. Under such conditions, our method becomes even more crucial for improving inference-time efficiency.
>
> In contrast, adding extra experts alone cannot fully resolve the straggler effect. It is challenging to accurately predict which experts will overflow in advance, and overloaded experts may still occur despite the increased expert count. Our method directly addresses the root cause: it is simpler, does not rely on repeatedly predicting overflowed experts, and guarantees that expert load stays below the capacity threshold after token dropping. Importantly, capacity-aware inference can be seamlessly combined with additional experts, providing a complementary and more robust solution.
>
> Thanks again for your comments. We will consider incorporating the above discussions in future revisions. At this stage, the manuscript space is limited, but we plan to provide a more complete discussion when preparing the next version.
>
> ---
>
> ### **W2: Discussion on the effect of different sequence lengths within a batch.**
>
> This is a good question. Since MoE layers flatten the input sequences into a single sequence before computation, the computational cost scales linearly with the total number of tokens. Load imbalance also becomes more influential when GPUs are highly utilized; otherwise, the GPUs have spare compute capacity and can maintain high throughput. Therefore, the results reported in the paper are measured under conditions of high GPU utilization.
>
> Regarding the ablation on sequence length, we conduct additional studies on DeepSeek-V2 here. Under a batch size of 8k, the measured end-to-end speedups are as follows:
>
> | input_SEQ | speedup |
> |-----------|---------|
> | 100       | 1.09    |
> | 200       | 1.18    |
> | 400       | 1.24    |
>
> This indicates that under heavier workloads, GPUs operate at higher utilization with less spare capacity, which makes the speedup more pronounced.
>
> ---
>
> ### **W3: The choice of local experts.**
>
> This depends on the total number of experts and GPUs, where \( m \) denotes the number of experts assigned per GPU (i.e., total experts divided by total GPUs). By default, we deploy experts sequentially across GPUs; for example, experts \(1 \ldots m\) are placed on GPU 1, and experts \(m+1 \ldots 2m\) are placed on GPU 2.
>
> Reordering experts is an interesting idea, but it introduces a very large optimization space, as each layer would have many possible permutations. We believe exploring such expert-placement strategies would be an interesting direction for future work.
>
> ---
>
> ### **W4: Prediction of performance after Token Drop or Expanded Drop.**
>
> Our results already demonstrate strong robustness across multimodal tasks. For these tasks, we recommend using more aggressive capacity factors, as image tokens typically exhibit substantial redundancy. In contrast, language tasks generally require higher capacity factors. Accurately predicting performance under capacity constraints is inherently challenging, much like forecasting performance after model compression. We believe this area remains under-explored and presents a promising direction for future work.
>
> Since routing decisions are driven by token representations rather than by representations of task names or task identifiers, it is difficult to explicitly link task-specific performance to the behavior of Token Drop or Expanded Drop. However, we outline several potential ideas that may offer insights. One direction is to introduce trainable dynamic capacity factors and use training-aware strategies to optimize the model under varying capacities, enabling flexible selection and improved performance. Another direction is to establish a knowledge base of validated tasks and compare incoming tasks or queries against this repository to estimate how small the capacity factor can be set without harming accuracy.
>
> ---
>
> Thanks for the insightful comments. We will incorporate these discussions into the updated draft.

---

### Author Response · Authors · 2025-12-03
**Summary of the Rebuttal and Revisions**

We sincerely thank the Area Chair and reviewers for their time, thoughtful feedback, and constructive suggestions during the discussion phase. We carefully addressed all raised points, and below we summarize the key responses and revisions we have provided in the rebuttal.

**Expanded empirical evaluation.**
To more thoroughly assess robustness, we incorporated additional experiments on long-context and generation-focused benchmarks in the revised submission, including HumanEval, NQ-Open, MTS-Dialog, and LongICLBench. These results consistently show that Token Drop and Expanded Drop remain stable across dynamic, multi-round, and long-sequence scenarios.

**Additional analysis for multi-expert-per-GPU regimes.**
We broadened our study by adding Mixtral results under a 2-expert-per-GPU configuration and introducing a device-level capacity variant. This variant yields stronger performance and higher speedups when multiple experts reside on the same GPU.

**Strengthened latency analysis.**
We also expanded our latency analysis with both layer-level and end-to-end measurements under varied workloads (different batch sizes and prompt lengths). These empirical findings reinforce the connection between expert load and MoE inference latency.

**Clearer algorithmic descriptions.**
In the response, we refined the explanations of Expanded Drop, the Image-First strategy, and the handling of rare cases where a token retains no experts after dropping. These clarifications were provided in the rebuttal to ensure the methodology is fully transparent and reproducible.

**Improved positioning with respect to related work.**
We clarified the distinction between our work and other inference optimization approaches. We highlighted that our method targets test-time load balancing, which an orthogonal bottleneck and can complement existing inference acceleration techniques.

**Refined writing and consistency.**
During discussion, we improved caption descriptions, clarified notation, fixed citation style, and polished several sections for clarity and readability. These updates were detailed in the rebuttal.

We appreciate the AC’s and reviewers’ efforts in helping us refine the work and are glad to provide any additional information if needed.

---

### Meta-Review · Area_Chair_G6m3 · 2026-01-13

**Summary:**

The paper addresses a practical and important problem: the straggler effect in MoE inference caused by imbalanced token-to-expert assignment. I find the proposed solution appealing because it is simple, training-free, and achieves up to 1.85× speedup with minimal performance loss across multiple architectures. One concern I initially had was whether this contribution is novel given existing load balancing work, but the authors convincingly demonstrated that training-time balancing does not carry over to inference, with even recent models like Qwen3-MoE exhibiting severe test-time imbalance. The additional experiments on generation and long-context tasks, along with device-level capacity constraints for multi-expert-per-GPU setups, further strengthen the contribution. I believe this work offers practical value to the community and merits publication. I recommend acceptance for this submission.

**Reviewer Concerns:**

The approach is simple and requires no training, achieving up to 1.85× speedup with minimal performance loss across multiple MoE architectures. One reviewer questioned novelty given existing load balancing work, but the authors showed that training-time balancing does not carry over to inference, with even recent models like Qwen3-MoE exhibiting severe test-time imbalance. The new experiments on generation and long-context tasks addressed concerns about token dropping affecting coherence. The device-level capacity constraints for multi-expert-per-GPU setups were added based on reviewer suggestions.

**Reviewer Scores:**

Initial scores were 8, 6, 4, 4. The two lower scores reflected concerns about novelty and limited workload analysis, both of which received substantial responses with new experiments. Given the thoroughness of the rebuttal and explicit openness from borderline reviewers, the consensus would likely support acceptance.

---

### Decision · Program_Chairs · 2026-01-26

Accept (Poster)